# Incorporating Hierarchical Semantics in Sparse Autoencoder Architectures

## Abstract

Sparse dictionary learning (and, in particular, sparse autoencoders) attempts to learn a set of human-understandable concepts that can explain variation on an abstract space. A basic limitation of this approach is that it neither exploits nor represents the semantic relationships between the learned concepts. In this paper, we introduce a modified SAE architecture that explicitly models a semantic hierarchy of concepts. Application of this architecture to the internal representations of large language models shows both that semantic hierarchy can be learned, and that doing so improves both reconstruction and interpretability. Additionally, the architecture leads to significant improvements in computational efficiency.

## 1 Introduction

Dictionary learning—and, in particular, sparse autoencoders (SAEs)—have attracted significant attention as a tool for interpreting representations in large language models (Bricken et al., 2023; Cunningham et al., 2023; Gao et al., 2024; Lieberum et al., 2024; Lindsey et al., 2025b). The aim of these methods is to jointly learn some set of human-understandable concepts that can explain the model's behavior, and a map from the model's internal representations to these concepts. Empirically, at least some of the features learned by SAEs do seem clearly semantically interpretable—for example, "Golden Gate Claude" used an SAE feature to coherently steer Claude (Templeton et al., 2024). However, despite considerable effort, these models have some strong limitations. In particular, the learned features generally do not suffice to accurately reconstruct the input—limiting the value for interpretability—and, as the model size increases, the features that are learned often seem semantically unnatural. For example, Chanin et al. (2024) find that increasing the model size leads to a phenomenon they call "feature splitting"—where a single high-level concept is represented by multiple low-level features. The underlying tension here is that pushing for accurate reconstruction leads us to increase the number of features, but increasing the number of features can lead to very specialized features that are not generally useful.

The goal of this paper is to improve this reconstruction-interpretability frontier by exploiting the hierarchical structure of semantics. The motivating observation is that concepts that are meaningful to humans are often organized in a hierarchical fashion—for example, corgi, greyhound, and shitzu are all particular instances of the general concept of dog. Standard dictionary learning will not make use of this hierarchical structure at all (instead, representing each feature individually). Intuitively, we would like to modify the data structure such that this hierarchy is explicitly represented. The hope is that this will allow us to capture the reconstruction power of large dictionaries while maintaining interpretability by appealing to the human-understandable structure of the hierarchy.

The main contribution of this paper is a new architecture for SAEs that explicitly, and highly efficiently, models hierarchical relationships between concepts. Concretely:

1. Park et al. (2024a) give foundational results on how hierarchical structure is represented in language models. We show how to translate this theory into a mixture-of-experts type architecture (see Figure 2) that captures hierarchical structure (see Figure 1).
2. We then show that this architecture strongly improves the reconstruction performance of SAEs, while maintaining or improving interpretability.

Figure 1: A Hierarchical Sparse Autoencoder architecture learns human interpretable hierarchy. Each box shows 5 of the strongest activating contexts for the feature, all tokens underlined activate the feature while the bold/text weight shows the relative strength of activation. For example, a "marriage" high-level feature and "divorce", "engagement", and "marriage" sublatents. Note that "48 sati svadba" is a Serbian wedding reality TV show and "Shaadi" is the Hindi word for marriage.

As an additional benefit, the new architecture is highly computationally efficient. In principle, this can allow scaling SAEs to much larger effective dictionary sizes, allowing fine-grained representations of a vast number of concepts.

## 2 BACKGROUND AND RELATED WORK

**Sparse Autoencoders** SAEs are a particular approach to sparse dictionary learning, a general class of unsupervised learning algorithms that aim to find a dictionary of human-interpretable features (or atoms) that can be used to reconstruct the input data. The hope is that by enforcing sparsity we can recover some set of underlying latent factors. There are a large number of methods based on this idea.

We'll focus on top-k SAE approach of Gao et al. (2024), which has the advantages of being simple and scalable. The idea is to map each input vector $x$ to a latent representation that is sparsified by a TopK operation that preserves the $k$ largest values and sets all others to zero. Then, the vector is reconstructed by decoding this sparse latent representation. In total, the SAE operation is given by:

$$\mathrm{SAE}_k(\mathbf{x}) = \mathbf{D}\,\mathrm{TopK}_k\big(\mathrm{LeakyReLU}_\alpha\left(\mathbf{E}\left(\mathbf{x} - \mathbf{b}\right)\right)\big) \tag{1}$$

where $\mathbf{x} \in \mathbb{R}^d$ is the input, $\mathbf{E}$ is the encoder matrix, $\mathbf{D}$ is the decoder matrix, $\mathbf{b}$ is a bias vector (with the rows, columns, and dimension of $\mathbf{E}$, $\mathbf{D}$, and $\mathbf{b}$ respectively being equal to the number of features) $k$ is a hyperparameter, and $\alpha = \frac{1}{\sqrt{d}}$ is an adaptive threshold for the LeakyReLU activation (where LeakyReLU has some small negative slope below the threshold). The vectors in $\mathbf{D}$ are referred to as "features" or "latent vectors." The parameters are learned by minimizing the reconstruction loss:

$$\mathcal{L}_{\mathrm{recon}} = \|\mathbf{x} - \mathrm{SAE}_k(\mathbf{x})\|_2^2 \tag{2}$$

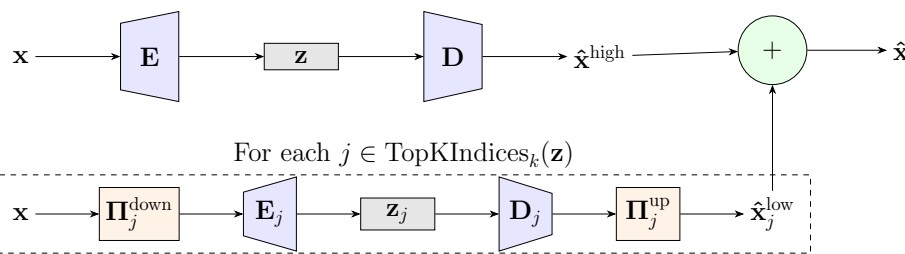

Figure 2: The structure of the Hierarchical Sparse Autoencoder architecture. The design combines a top-level encoder-decoder (upper path) that captures general concepts with expert-specific autoencoders (lower paths) that model refined features. For a given input, only a sparse subset of experts is activated, enhancing computational efficiency while maintaining expressive power.

**Related Work**  The use of sparse autoencoders in LLM interpretability has been a topic of considerable interest (Marks et al., 2025), (Lindsey et al., 2025a), (Engels et al., 2024), (Kissane et al., 2024).

The most closely related is a line of work focused on ameliorating feature splitting. For instance, Matryoshka SAEs (Bussmann et al., 2025) and EWG-SAEs (Li & Ren, 2025) both operate by defining groupings of dictionary atoms and then modifying the training objective of the SAE to encourage some of the groupings to contain coarser features, and other groupings to contain finer features. These approaches are also motivated by the hierarchical nature of semantics. In contrast to the approach we take here, they do not modify the flat set structure of the architecture. Modifying the architecture has the advantages that it both makes the hierarchical structure explicit—improving interpretability—and also allows us to leverage the hierarchical structure to greatly improve computational efficiency, as we will see.

We also highlight Switch SAEs (Mudide et al., 2024), which introduce a mixture-of-experts type routing mechanism shares some structural similarities to our approach. However, this approach is entirely motivated by pushing the reconstruction quality vs sparsity frontier for a given compute budget. Each expert has no high vs low-level distinction and isn't designed to contain a set of features that should be interpreted together. Indeed, they find that their decoder features do not cluster in any particular way.

The ideas here connect to the broader field of causal representation learning (CRL), which aims to learn the causal factors underlying a data-generating process (Locatello et al., 2019; 2020; Schölkopf et al., 2021; Ahuja et al., 2022a; Brehmer et al., 2022; Lippe et al., 2023; Moran & Aragam, 2025). Recent work has extended CRL to foundation model analysis, seeking identifiable and disentangled representations of concepts (Rajendran et al., 2024; Joshi et al., 2025). In this spirit, we also aim to learn disentangled concept representations.

## 3  HIERARCHICAL SPARSE AUTOENCODER ARCHITECTURE

We now turn to the development of an architecture that bakes in the hierarchical structure of concepts.

### 3.1  ARCHITECTURE

**Hierarchical Geometry**  Our main inspiration follows Park et al. (2024a), who find that the representations of categorical concepts in language models have a specific geometric structure. In particular, they show that every categorical concept has *two* representations: a parent feature indicating whether the concept is active, and a low-rank subspace of the representation space containing a polytope where each point is a child features corresponding to a different possible value of the concept. For example, the concept "dog" is represented by a parent feature indicating whether the concept is active ('is dog' vs 'not dog'), and a low-dimensional subspace containing a polytope where each point is a child feature corresponding to a different breed of dog.

The idea is to create an architecture that matches this dual representation structure; see Figure 2. To achieve this, we propose an architecture with three parts:

1. A top-level SAE that aims to capture the binary feature indicating whether a high-level concept is active. This is a standard SAE with a relatively small number of features.
2. For each atom in the high-level SAE, we have an associated down projector onto a low-dimensional subspace (and an up-projector mapping back to the original space). This corresponds to the low-dimensional subspace associated to the categorical concept. Since each high-level concept spans a low-dimensional subspace, the only requirement for the 'projection' is staying within the column space of the concept. Thus, we do not impose idempotence or symmetry and simply use 2 trainable matrices.
3. For each atom in the high-level SAE, we have a low-level SAE that operates on the low-dimensional subspace associated with atom. The elements of each such SAE correspond to the different possible values of the categorical concept (e.g., the different breeds of dog).

That is, the architecture is a high-level SAE where each atom also has a low-level SAE attached to it.

Now, a critical observation here is that an atom in the low-level SAE can only be active if the high-level atom is also active. That is, for the residual stream vector $x$ to encode the concept of "corgi" it *must* also encode the concept of "dog." To respect this constraint, we structure the model as a mixture-of-experts type architecture, where a low-level SAE is only called if the associated high-level feature is active. This both respects the hierarchical structure of the concepts and also allows for an enormous computational efficiency gain (see below).

In fact, the Park et al. (2024a) results are more precise than the informal presentation above. In particular, they show that a low level concept ("corgi") is naturally represented as the *sum* of a vector for the high-level concept ("dog") and a vector representing the low-level concept in the context of the high level space (i.e., "corgi" = "dog" + "corgi | dog"). It is these contextual representations that live in a low-dimensional space. This overall structure—a low level concept is represented as the sum of a high-level concept plus the low-level concept in the context of the high-level one—is exactly the structure implemented by the H-SAE.

**Forward Pass**    We can now make the architecture precise:

$$\text{H-SAE}(\mathbf{x}) = \sum_{j \in \text{TopKIndices}_k} \big( \underbrace{z_j \mathbf{d_j}}_{\hat{\mathbf{x}}^{\text{high}}} + \underbrace{\mathbf{\Pi}_{\mathbf{j}}^{\text{up}} \text{SAE}_1^j(\mathbf{\Pi}_{\mathbf{j}}^{\text{down}}\mathbf{x})}_{\hat{\mathbf{x}}_j^{\text{low}}} \big) \tag{3}$$

where $\text{TopKIndices}_k$ are the indices of the top $k$ features, $\mathbf{d_j}$ is the $j$-th high-level feature, $z_j = (\text{LeakyReLU}(\mathbf{Ex}))_j$ is the corresponding code, $\text{SAE}_1^j$ is the expert-specific autoencoder for high-level feature $j$, and $\mathbf{\Pi}_{\text{up}}^{\mathbf{j}}$ and $\mathbf{\Pi}_{\text{down}}^{\mathbf{j}}$ are projection matrices (of dimension $s$) that map the input to the expert subspace and back to the original space, respectively. Note that the low-level SAE uses $\text{TopK}_1$ over its $a$ features, retaining only a single low-level feature for each expert. This corresponds to the idea that the representation of a subordinate concept should be at a particular vertex in the polytope corresponding to the categorical concept.

We depict this architecture visually in Figure 2 and algorithmically in Algorithm 1.

**Computational Efficiency**    Beyond explicitly enforcing the target semantic structure, activating the low-level SAE only when the corresponding high-level feature is active provides a significant computational advantage. The computational cost of a forward pass can be expressed as follows:

$$\text{Cost}_{\text{forward}} = \underbrace{O(jd)}_{\text{High-level encoding}} + \underbrace{O(ksd)}_{\text{Subspace projection}} + \underbrace{O(kas)}_{\text{Low-level encoding}}$$
$$+ \underbrace{O(kas)}_{\text{Low-level decoding}} + \underbrace{O(ksd)}_{\text{Upward projection}} + \underbrace{O(kd)}_{\text{High-level decoding}}$$

where $j$ is the number of high-level features (experts), $d$ is the dimensionality of the input activation vector, $s$ is the subspace dimension for each expert, $a$ is the number of low-level features per expert, and $k$ is the sparsity level (number of activated experts). In the typical case where $k \ll j$ (sparsity) and $s \ll d$ (low dimensional subspace) the cost is dominated by the top-level SAE. That is, equipping

---

**Algorithm 1** Hierarchical SAE Forward Pass and Loss Computation

---

**function** ForwardPass($\mathbf{x}$)
    $\mathbf{z} \leftarrow \text{TopK}_k(\text{LeakyReLU}_\alpha(\mathbf{Ex}))$                          ▷ High-level encoding
    $\mathcal{K} \leftarrow$ indices of nonzero elements in $\mathbf{z}$
    $\hat{\mathbf{x}}^{\text{high}} \leftarrow \mathbf{Dz}$                                    ▷ High-level reconstruction
    $\hat{\mathbf{x}}^{\text{low}} \leftarrow \mathbf{0}$                                   ▷ Initialize low-level reconstruction
    **for** $j \in \mathcal{K}$ **do**                               ▷ Only process activated experts
        $\mathbf{x}_j^{\text{sub}} \leftarrow \mathbf{\Pi}_j^{\text{down}}\mathbf{x}$                       ▷ Project to expert subspace
        $\mathbf{z}_j \leftarrow \text{LeakyReLU}_\alpha(\mathbf{E}_j\mathbf{x}_j^{\text{sub}})$               ▷ Low-level encoding
        $\hat{\mathbf{x}}_j^{\text{sub}} \leftarrow \mathbf{D}_j\mathbf{z}_j$                       ▷ Reconstruct in subspace
        $\hat{\mathbf{x}}^{\text{low}} \leftarrow \hat{\mathbf{x}}^{\text{low}} + \mathbf{\Pi}_j^{\text{up}}\hat{\mathbf{x}}_j^{\text{sub}}$        ▷ Project back and accumulate
    **end for**
    $\hat{\mathbf{x}} \leftarrow \hat{\mathbf{x}}^{\text{high}} + \hat{\mathbf{x}}^{\text{low}}$                         ▷ Combined reconstruction
    **return** $\hat{\mathbf{x}}, \mathbf{z}, \{\mathbf{z}_j\}_{j \in \mathcal{K}}$
**end function**

**function** ComputeLoss($\mathbf{x}, \hat{\mathbf{x}}, \mathbf{z}, \{\mathbf{z}_j\}_{j \in \mathcal{K}}$)
    $\mathcal{L}_{\text{recon}} \leftarrow \|\mathbf{x} - \hat{\mathbf{x}}\|_2^2$
    $\mathcal{L}_{\text{top\_recon}} \leftarrow \|\mathbf{x} - \mathbf{Dz}\|_2^2$
    $\mathcal{L}_{\text{recon}} \leftarrow \mathcal{L}_{\text{recon}} + \beta\mathcal{L}_{\text{top\_recon}}$         ▷ Encourage meaningful top-level features
    $\mathcal{L}_{\text{sparse}} \leftarrow \|\mathbf{z}\|_1 + \sum_{j \in \mathcal{K}} \|\mathbf{z}_j\|_1$
    $\mathcal{L}_{\text{ortho}} \leftarrow \frac{\|\mathbf{DE} - \text{diag}(\mathbf{DE})\|_F}{n^2 - n}$
    $\mathcal{L} \leftarrow \mathcal{L}_{\text{recon}} + \lambda_1\mathcal{L}_{\text{ortho}} + \lambda_2\mathcal{L}_{\text{sparse}}$
    **return** $\mathcal{L}$
**end function**

---

the top-level SAE with hierarchical structure adds negligible computational overhead. Because the hierarchical SAE is much more expressive, this significantly improves SAE scalability. We also note that because the memory cost of a batch gradient step scales with the number of activated parameters, not the total number of parameters, this efficiency also applies to (per-step) training. Indeed, in practice we find that the additional cost imposed by the hierarchical structure is very small.

## 3.2 TRAINING OBJECTIVE

We also make some minor modifications to the training objective, using the following loss function:

$$\mathcal{L} = \mathcal{L}_{\text{recon}} + \lambda_1\mathcal{L}_{\text{ortho}} + \lambda_2\mathcal{L}_{\text{sparse}} \tag{4}$$

**Reconstruction Loss** Following standard practice, we measure reconstruction loss with Euclidean distance. Additionally, to encourage the top-level SAE features to be meaningful in their own right (rather than just as routers) we also penalize reconstruction error from the top-level SAE only. The reconstruction loss is then:

$$\mathcal{L}_{\text{recon}} = \|\mathbf{x} - \text{H-SAE}(\mathbf{x})\|_2^2 + \beta\|\mathbf{x} - \hat{\mathbf{x}}^{\text{high}}\|_2^2, \tag{5}$$

where $\beta = 0.1$ was chosen because the top-level SAE alone has less capacity than the full H-SAE.

**Auxiliary Losses** Park et al. (2024b) show that the representations of "causally separable" (individually manipulable) features are bi-orthogonal in a certain sense.[1] This suggests that we could discourage semantic redundancy in the learned features (e.g., having multiple features for "dog") by imposing a suitable orthogonality constraint on the learned features. To operationalize this, we introduce a bi-orthogonality penalty on the top-level features:

$$\mathcal{L}_{\text{ortho}} = \frac{\|\mathbf{DE} - \text{diag}(\mathbf{DE})\|_F}{n^2 - n}, \tag{6}$$

---

[1]Namely, the dot product between primal and dual space representations is zero.

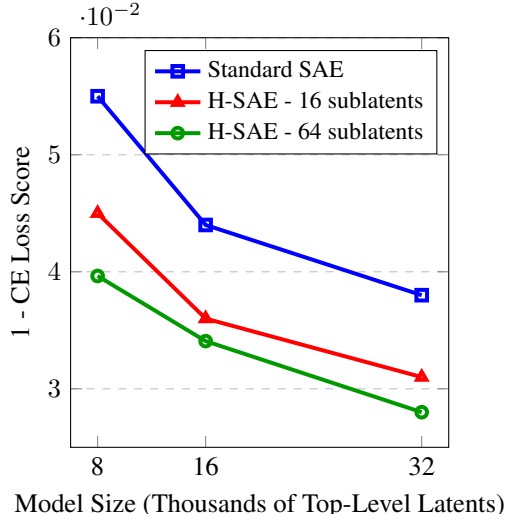

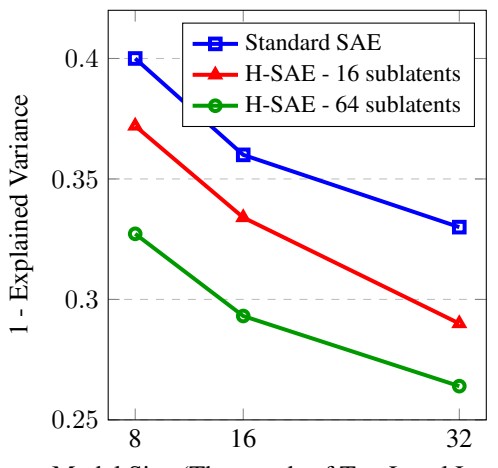

(a) The H-SAE has better reconstruction performance than a standard SAE as measured 1 - CE Loss score across different model sizes and with both 16 and 64 sublatents per expert. Lower values indicate better downstream model performance, using the SAEBench CE loss score.

(b) The H-SAE has better reconstruction performance than a standard SAE as measured 1 - Explained Variance across different model sizes and with both 16 and 64 sublatents per expert. Lower values indicate better capture of the data's inherent structure.

Figure 3: H-SAEs have better reconstruction performance as measured by explained variance and the downstream CE loss of a Gemma 2-2B using SAE-reconstructed activation vectors.

where $\text{diag}(\mathbf{DE})$ is the diagonal matrix formed by the diagonal elements of the top-level decoder multiplied by the top-level encoder, and $\|\cdot\|_F$ denotes the Frobenius norm. This pushes the encoder representation for feature $i$ and decoder representation for feature $j$ to be orthogonal if $i \neq j$. Mechanically, this means that if we ran the encoder on the autoencoder's own output, whether feature $z_i$ was active in the first encoding would not effect feature $z_j$ in the second encoding.

The motivation here is to encourage compositionality in the learned features. However, it is not clear how to empirically measure compositionality, and so we are unsure whether this term actually achieves this goal. Nevertheless, we do observe empirically that adding this term does an excellent job of mitigating the "dead atom" phenomena where many features are never used in reconstructions. We use this instead of the auxiliary dead latent loss proposed by Gao et al. (2024), but we do not believe this is a crucial component. See Section B for ablations.

We also include a small $\ell_1$ sparsity penalty on the latent values on both the top and low-level features outside the top k to further encourage specialization; this is the $\mathcal{L}_{\text{sparse}}$ term. Again, it's unclear how to measure the target compositionality effect, and we do not believe this is a key component. See Section B for ablations.

## 4 EXPERIMENTS

The main motivation for this work is that by incorporating the hierarchical structure of semantics we can improve the reconstruction-interpretability frontier. Accordingly, we want to answer three questions:

1. Does the H-SAE improve reconstruction?
2. Does the H-SAE maintain, or improve, interpretability?
3. Does the model in fact learn hierarchical semantics?

We will see that the answer to all three questions is yes.

**Experimental Setup** Our training data consists of 1 billion residual stream vectors extracted from layer 20 of Gemma 2-2B . We collect this data by running Gemma on a large corpus of Wikipedia

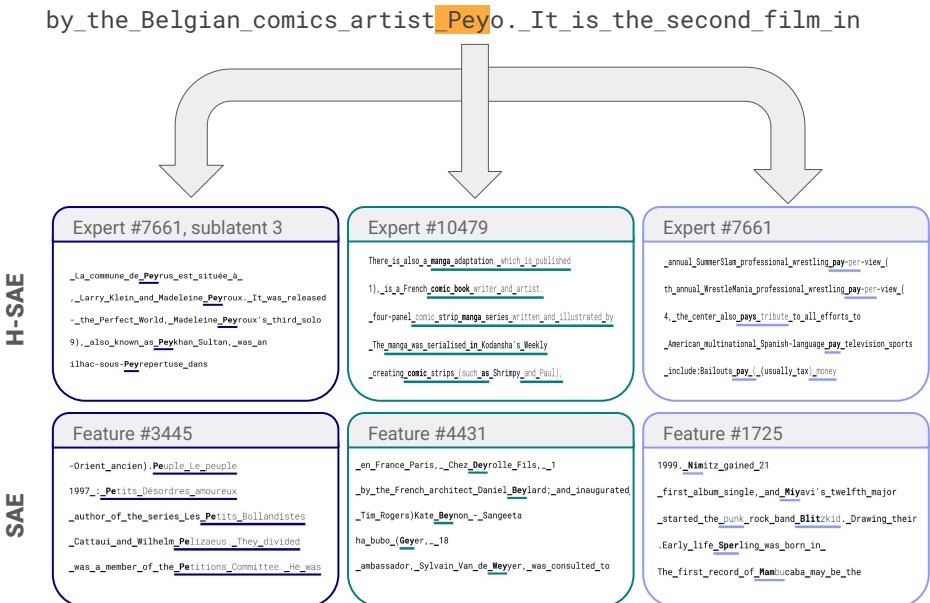

Figure 4: A comparison decomposition of the same context/embedding in the H-SAE architecture and standard SAE. The H-SAE represents a token about the creator of the Smurfs cartoon using a high level "Pay" homophone feature, and a low-level "Pey" token feature. This feature is then composed with a comic books feature. Wheras, the SAE has features for "words that end with 'ey'" and "Pe" words, but none of the top-32 contain a high-level feature for the 'comic book' context of this sentence, rather the SAE uses a 'part of an artists' name feature to reconstruct the embedding.

articles, spanning multiple languages and a wide range of topics. For each article, we extract the residual stream vectors corresponding to the first 256 tokens. Following standard practice, we normalize the vectors to unit norm (Lieberum et al., 2024). However, we do not subtract any mean vector nor include a bias term as Gao et al. (2024) do, as we found this decreased training stability.

As a baseline, we use the TopK SAE architecture (Gao et al., 2024). The H-SAE is trained using the objective function described in Equation (4). All models are trained on the same data for 4 epochs. The baseline top-k autoencoders empirically perform equivalently on reconstruction loss to the Gemma Scope JumpReLU autoencoders (Lieberum et al., 2024). See Section A for more details on training.

**Reconstruction**   Figure 3 shows the reconstruction performance of the H-SAE and the baseline at a variety of dictionary sizes. We measure both 1 - explained variance (i.e. normalized $\ell^2$ reconstruction loss) and the LLM CrossEntropy loss induced by replacing the original activations with the reconstructions (normalized by SAEBench between 0 and 1). As expected, adding hierarchical capacity to the model improves reconstruction performance. Indeed, this effect is dramatic. The H-SAE with 64 sublatents per expert is on par with the standard SAE with 32 top-level features, despite having $1/4$ the compute cost.

**Matryoshka SAE**   Matryoshka SAEs (Bussmann et al., 2025) are the most similar SAE architecture conceptually, hoping to reduce undesirable feature absorption and increase hierarchical interpretability. To compare these models, we start by training them on the token unembeddings, as in other ablations. The features are fairly similar, with the H-SAE often finding sparser representations, finding high-level features the M-SAE does not, and fewer uninterpretable features. These factors also combine with the reconstruction loss and computational advantage the H-SAE has over both an M-SAE and standard SAE. Additionally, we train both an H-SAE and M-SAE on the synthetic benchmark from the M-SAE paper, and find the H-SAE outperforms the M-SAE. Full results and experimental details are in Section D.

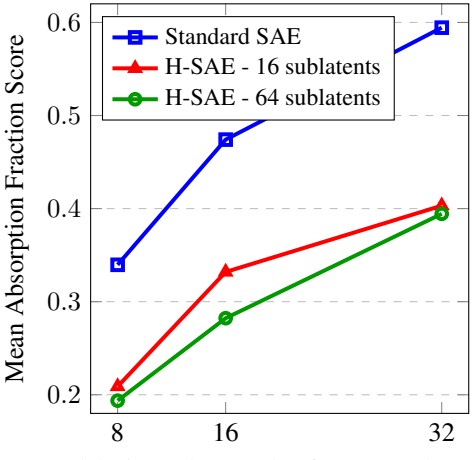

| Language Pair | H-SAE | SAE |
|---|---|---|
| English vs French | **8.448** | 9.772 |
| English vs Spanish | **8.190** | 9.598 |
| English vs German | **9.372** | 10.938 |
| French vs Spanish | **5.748** | 7.170 |
| French vs German | **7.306** | 9.038 |
| Spanish vs German | **7.476** | 9.270 |

(a) The standard SAE shows higher absorption, indicating greater feature splitting, while our H-SAE architecture maintains more coherent representations as measured by the mean fraction of first-letter classifications that show signs of absorption.

(b) The H-SAE architecture has less divergence in features for the same sentence in different languages. Higher values indicate more redundant features, as measured by the mean set difference between the same token in different languages, demonstrating our H-SAE architecture's superior ability to learn highly composable features.

Figure 5: H-SAEs exhibit less undesirable feature absorption and fewer redundant features across languages

## 4.1 INTERPRETABILITY

By itself, an improvement in reconstruction performance may not be meaningful. The reason is that there are many possible modifications that improve reconstruction performance by sacrificing interpretability. This concern is somewhat mitigated by the fact that the extra capacity of the H-SAE is highly constrained; low-level features can only be activated when their corresponding high-level feature is active. Nevertheless, we would like to directly evaluate the interpretability of the H-SAE against a standard SAE.

Figure 4 shows a comparison of the baseline and H-SAE decomposition of the same context. We observe that the H-SAE has features that are of the same quality, if not better. This behavior is typical. See Section C for more examples of features and supplementary materials for an interactive notebook including visualizations of hundreds of features from each model.

To complement the visualizations, we perform some more systematic tests. Recent work has shown that general-purpose automated interpretability evaluations are misleading–particularly regarding more abstract features (Heap et al., 2025). Accordingly, we focus in particular on feature splitting and absorption. These are aspects that we would expect to see the greatest effect on through learning more general top-level features, and are relatively measurable. To test feature absorption, we run the first letter classification benchmark from SAEBench (Karvonen et al., 2025). This benchmark tests the improvement in probing performance for a first letter classification task as the number of features used increases above 1. This task should only require a single feature if there is no undesirable splitting or absorption. As expected, we find that the H-SAE architecture both improves performance on this task and decreases the rate of increase in absorption as width increases. See Figure 5a, where we report the fraction of the projection from SAE activations onto a first-letter classification probe that is not explained by a single feature.

We can also test for the presence of duplicate features or 'redundancy'. One particular example of this that we and others observe is equivalent syntactic features from different languages (e.g. a French period and an English period) (Lindsey et al., 2025b). So, we sample 1,000 English sequences from the training data, consisting of between 16 and 48 tokens. We then use Gemini-2.0-Flash to translate these sentences into French, Spanish, and German (the most common non-English languages in the training data). Then, we collect the top 8 most strongly activated features on the last token of the context for the standard and H-SAE. Ideally, the features activated by the same token in different

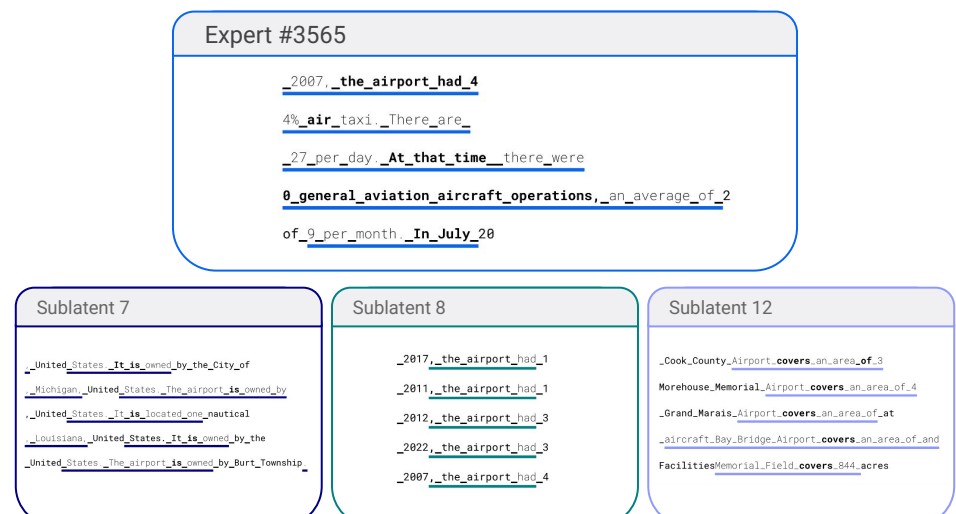

Figure 6: The H-SAE has a "airports" high-level feature and "US airport", "the token 'airport'", and "airport size" sublatents.

languages should be similar. We measure the size of the symmetric set difference between the top 8 features activated by the same token in different languages (a value between 0 and 16); see Figure 5b. As expected, the H-SAE uses more similar sets of features across all language pairs than the baseline SAEs.

**Hierarchical Semantics**    Finally, we turn to the question of whether the H-SAE is indeed learning hierarchical semantics. To this end, we visualize examples where low-level features are strongly activated. Figures 1, 4 and 6 show examples from the 16k experts x 16 sublatents H-SAE. We observe that hierarchical semantics are clearly emerging. See the supplementary materials for an interactive notebook including hundreds of such visualizations.

## 5    DISCUSSION

The main question in this work is whether the interpretability-reconstruction frontier can be improved by exploiting the hierarchical structure of semantics. We find that the answer is yes. Incorporating hierarchy dramatically improves reconstruction and moderately improves interpretability of the top-level features. Further, the two-level structure effectively communicates semantic relationships between concepts—e.g., we observed high-level latents capturing regional concepts like "Bay Area," with corresponding expert-specific sub-latents representing entities like "Stanford" that exist within that region. As a further benefit, the hierarchical structure allows for large increases in the computational efficiency relative to the effective number of atoms.

**Limitations and Further Work**    In this work we primarily analyze the architectural changes in the context of a simple 'standard' SAE approach. Incorporating hierarchy leads to an improvement, but the results are still imperfect—e.g., we still find some level hard-to-interpret features, and we still far short of perfect reconstruction. It is unclear whether this reflects a fundamental limitation of dictionary learning, or whether there is some additional set of changes that would lead to dramatically improved performance. For example, our ablation studies on word unembeddings suggest that using a reconstruction objective other than Euclidean distance can massively improve interpretability; see Section B.3. Similarly, work in causal representation learning has shown simple sparsity objectives have fundamental limitations (Locatello et al., 2019), and developed a variety of more sophisticated objectives leading to much better performance (Locatello et al., 2020; Ahuja et al., 2022b; Lippe et al., 2022). It would be exciting to find ways to incorporate such insights into LLM interpretability.

## REPRODUCABILITY

All the main experiments and results are reproducible using the code and data provided in the supplementary material. The full dataset used for training is too large to include directly as it is 10TB, but the full code used to create it is included. Model checkpoints are provided for all H-SAEs and standard SAEs trained. To aid with evaluation, supplementary material also includes hundreds of randomly selected features to demonstrate the consistent interpretability and quality of the learned features.

## ACKNOWLEDGEMENTS

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

## A  TRAINING DETAILS

### A.1  INPUT DATA CONSTRUCTION

- The 20231101 Wikipedia dump is used to construct the data. Approximately 500 million English tokens are selected by taking the first 256 tokens among articles on English and Simple English Wikipedia. The articles are selected by choosing the top 2 million longest articles (using length as a proxy for quality to filter out low quality stub articles) and then randomly among those articles until 500 million tokens have been selected.

- For non-English tokens, articles are chosen randomly from the largest 128 non-English Wikipedias (as measured by active users). The mix of languages is also proportional to the number of active users. This is used as a rough proxy for Wikipedia quality.

- Tokens are then packed into sequences to fill the Gemma 2-2B context window and activations are collected using Penzai and stored on disk with Tensorstore (Johnson, 2024; Maitin-Shepard & Leavitt, 2022).

- BOS and EOS tokens are stripped from the data and shuffled on disk prior to training. SAE training requires data access well in excess of 1GB/s to saturate a 8xA100 node and so the data must be shuffled ahead of time.

### A.2  MODEL IMPLEMENTATION AND HYPERPARAMETERS

- The hierarchical sparse autoencoder is implemented in JAX and Equinox (Bradbury et al., 2018; Kidger & Garcia, 2021).

- We train with a batch size of 32,512 and top-k of 32. When the number of sublatents per expert is 16, the subspace dimension is 4. Otherwise, it is 8.

- We set the orthogonality penalty and top-level reconstruction coefficients to $0.1$. The l1 coefficient is $0.001$. For standard SAEs that require an auxiliary loss to prevent dead latents we use a coefficient of $1/30$.

- We use the adam optimizer with global norm clipping of 0.75 and b1 of 0.9.

- The learning rate is $5 \cdot 10^{-4}$, initialized to $10^{-11}$ with a 1,000 steps linear warmup and cosine decay over the remainder of the training run.

- Additionally, the regularizers also warmup from 0 over the first 1,000 steps.

### A.3 TRAINING DYNAMICS

- We track an exponential moving average over 300 batches (i.e. roughly every million tokens) of the number of times latents activate. Both the auxillary loss and orthogonality loss push the number of latents that don't activate in 1 million tokens (i.e. dead) well under 1%. The dead atom auxillary loss coincidentally has exactly the same compute cost as the addition of hierarchy with the hyperparameters used for experiments. However, in terms of wall-time the H-SAE actually trains slightly faster due to the auxillary loss being a much more memory-intensive operation using a naive implementation. We do not attempt to use maximally efficient implementations so detailed compute analysis is not conducted.

## B ABLATIONS

### B.1 PRE-PROCESSING

We conduct ablations by evaluating our H-SAE architecture applied to the token unembedding matrix of the Gemma 2-2B model, rather than internal activations, as the unembedding matrix provides a small data set that is more suitable for ablation studies. Gemma 2-2B has 256,128 tokens in its vocabulary, however many of these are relatively rare and meaningless tokens. After discarding these, we are left with 186,032 tokens.

Previous work (Park et al., 2024b) has shown that the semantic structure of the unembedding matrix is fundamentally non-Euclidean, meaning that the standard Euclidean inner product does not align orthogonality with semantic independence. Following this line of work, we whiten the unembedding matrix by multiplying it by the inverse square root of the covariance matrix, which has been shown to align the inner product with the underlying geometry of the data. Empirically, we observe that this whitening step is necessary for the model to learn meaningful features.

### B.2 ABLATION SETUP

For the ablations, we apply our H-SAE architecture to this whitened matrix, using a top-level dictionary with 2048 latent vectors (experts) and 32 sublatents per expert, $k = 5$. We considered an $\ell_1$ strength of $2.5 \times 10^{-3}$ and an orthogonality penalty of $2.5 \times 10^{-2}$. As our focus was on ablations, we were not concerned with the absolute performance of the model, but rather with the relative performance of different configurations, so we did not tune hyperparameters beyond those being compared. We trained for $10,000$ steps with a batch size of 8192 and a cosine-decay learning rate schedule starting at $8192 \times 10^{-15}$ and peak value of $8192 \times 10^{-6}$.

We are interested in whether the orthogonality loss and $\ell_1$ regularizer are necessary for interpretability and hierarchy. Because this is fundamentally a qualitative question, we do not report quantitative results. Instead, we qualitatively analyze the learned features and their hierarchical structure by examining the top activated features for a set of candidate tokens (e.g., "puppy"). We analyze these features in the same way as we do for the embeddings, examining the max-activating examples for each feature.

We test the orthogonality loss with and without the $\ell_1$ regularizer, and find that the orthogonality loss does not seem to be necessary for interpretability and hierarchy. Nonetheless, we chose to keep it in our final runs on the embeddings as it helps reduce the number of dead latents, which is a practical concern.

We also test the $\ell_1$ regularizer with and without the orthogonality loss, and find that it too does not seem to be necessary for interpretability and hierarchy. Ultimately, we chose to keep it for our final runs on the embeddings to allow for some adaptivity beyond the fixed top-k selection, as we have observed in practice that very low activations are often not meaningful, so we would like to encourage the model to use only the most informative features.

Displayed below in Figures 7, 8, 9, 10, 11, 12, and 13 are the example results of our ablations studies on the unembedding matrix. These were not chosen randomly but rather to show a variety of different types of features.

```
Explanations for word: 'puppy'
================================================

◆ Top-Level Feature 732 (Activation: 0.3964)
  Words that maximally activate this feature:
  [' dog', ' Dog', 'Dog', 'dog', ' dogs']
    ↳ Low-Level Feature: 31 (Activation: -0.00061521)
      Words that maximally activate this low-level feature:
      [' barks', ' собаки', ' woof', ' собака', ' Labrador']

◆ Top-Level Feature 956 (Activation: 0.0883)
  Words that maximally activate this feature:
  [' pig', ' goat', ' pigs', ' monkey', ' Goat']
    ↳ Low-Level Feature: 12 (Activation: 0.00009925)
      Words that maximally activate this low-level feature:
      [' horse', 'Horse', ' Horse', ' pony', ' Pony']

◆ Top-Level Feature 117 (Activation: 0.0525)
  Words that maximally activate this feature:
  [' boy', ' Boy', 'boy', ' Boy', ' BOY']
    ↳ Low-Level Feature: 0 (Activation: 0.21417145)
      Words that maximally activate this low-level feature:
      [' menina', ' boya', 'ガール', 'BOYS', 'GIRLS']

◆ Top-Level Feature 487 (Activation: 0.0457)
  Words that maximally activate this feature:
  [' pet', ' Pet', 'pet', 'Pet', ' PET']
    ↳ Low-Level Feature: 4 (Activation: 0.00009525)
      Words that maximally activate this low-level feature:
      ['PETER', ' Pedro', 'Pedro', ' petrol', ' Petrol']

◆ Top-Level Feature 1809 (Activation: 0.0437)
  Words that maximally activate this feature:
  [' patient', ' shopper', ' attendee', ' subscriber', ' sufferer']
    ↳ Low-Level Feature: 8 (Activation: 0.00009030)
      Words that maximally activate this low-level feature:
      [' Passenger', ' passenger', 'passenger', 'Passenger', ' guest']
```

(a) Baseline

```
Explanations for word: 'puppy'
================================================

◆ Top-Level Feature 1443 (Activation: 0.3872)
  Words that maximally activate this feature:
  [' dog', ' Dog', ' dogs', 'dog', 'Dog']
    ↳ Low-Level Feature: 3 (Activation: 0.00008098)
      Words that maximally activate this low-level feature:
      [' pup', ' Pup', ' pups', 'puppy', ' Puppy']

◆ Top-Level Feature 1878 (Activation: 0.2606)
  Words that maximally activate this feature:
  [' baby', ' Baby', 'baby', 'Baby', ' babies']
    ↳ Low-Level Feature: 31 (Activation: 0.00002946)
      Words that maximally activate this low-level feature:
      [' babys', ' diapers', ' diaper', ' Babys', ' pram']

◆ Top-Level Feature 1530 (Activation: 0.0658)
  Words that maximally activate this feature:
  [' novice', ' beginner', ' newbie', ' rookie', ' beginners']
    ↳ Low-Level Feature: 23 (Activation: 0.00004928)
      Words that maximally activate this low-level feature:
      [' beginner', ' Beginner', 'Beginner', ' Beginners', ' beginners']

◆ Top-Level Feature 1783 (Activation: 0.0564)
  Words that maximally activate this feature:
  [' Bad', 'Bad', ' bad', 'bad', ' BAD']
    ↳ Low-Level Feature: 3 (Activation: 0.00006972)
      Words that maximally activate this low-level feature:
      [' Terrible', '徽', ' mauvais', '不好的', ' pobres']

◆ Top-Level Feature 1055 (Activation: 0.0516)
  Words that maximally activate this feature:
  [' boy', 'boy', ' Boy', 'Boy', ' guy']
    ↳ Low-Level Feature: 12 (Activation: 0.00003262)
      Words that maximally activate this low-level feature:
      ['BOYS', ' Gentleman', ' chàng', ' Jungen', ' GUYS']
```

(b) No Orthogonality

```
Explanations for word: 'puppy'
================================================

◆ Top-Level Feature 658 (Activation: 0.4604)
  Words that maximally activate this feature:
  [' dog', ' Dog', ' dogs', 'Dog', 'dog']
    ↳ Low-Level Feature: 28 (Activation: -0.00005854)
      Words that maximally activate this low-level feature:
      ['🐕', '狗狗', ' собак', ' щен', ' cão']

◆ Top-Level Feature 214 (Activation: 0.1276)
  Words that maximally activate this feature:
  [' beginner', ' novice', ' beginners', ' rookie', ' newbie']
    ↳ Low-Level Feature: 27 (Activation: 0.00003545)
      Words that maximally activate this low-level feature:
      [' aspiring', '新人', ' amateur', ' Amateur', ' amateurs']

◆ Top-Level Feature 1986 (Activation: 0.0730)
  Words that maximally activate this feature:
  [' teen', ' teens', ' Teen', 'Teen', ' Teens']
    ↳ Low-Level Feature: 5 (Activation: 0.00006836)
      Words that maximally activate this low-level feature:
      ['teens', ' teen', ' Jugend', ' Adolescent', ' Adoles']

◆ Top-Level Feature 175 (Activation: 0.0657)
  Words that maximally activate this feature:
  [' tiger', ' Tiger', ' tigers', 'Tiger', ' giraffe']
    ↳ Low-Level Feature: 0 (Activation: 0.00009033)
      Words that maximally activate this low-level feature:
      [' bunny', ' Rabbit', ' squirrel', ' Hedgehog', ' Butterfly']

◆ Top-Level Feature 818 (Activation: 0.0502)
  Words that maximally activate this feature:
  [' cow', ' cows', ' Cow', 'cow', 'Cow']
    ↳ Low-Level Feature: 11 (Activation: -0.00002557)
      Words that maximally activate this low-level feature:
      [' horse', ' Horse', 'Horse', ' donkey', 'donkey']
```

(c) No L1

```
Explanations for word: 'puppy'
================================================

◆ Top-Level Feature 1354 (Activation: 0.3591)
  Words that maximally activate this feature:
  [' dog', ' Dog', 'Dog', ' dogs']
    ↳ Low-Level Feature: 20 (Activation: -0.00033513)
      Words that maximally activate this low-level feature:
      ['hound', 'Hound', 'hounds', ' Hound', ' hound']

◆ Top-Level Feature 166 (Activation: 0.0785)
  Words that maximally activate this feature:
  [' pregnancy', ' Pregnancy', ' pregnant', ' pregnancy', ' pregnancies']
    ↳ Low-Level Feature: 10 (Activation: 0.19668788)
      Words that maximally activate this low-level feature:
      [' abort', ' abortion', ' abortions', ' Abortion', ' breastfeeding']

◆ Top-Level Feature 1866 (Activation: 0.0374)
  Words that maximally activate this feature:
  [' tiger', ' Tiger', ' lion', ' Lion', ' wolf']
    ↳ Low-Level Feature: 9 (Activation: 0.00006496)
      Words that maximally activate this low-level feature:
      [' chimpanze', ' lions', ' Bunny', ' wolf', ' Wolves']

◆ Top-Level Feature 239 (Activation: 0.0317)
  Words that maximally activate this feature:
  [' card', ' cards', ' Card', ' Cards', 'card']
    ↳ Low-Level Feature: 1 (Activation: 0.00007931)
      Words that maximally activate this low-level feature:
      ['cartão', 'Karten', 'Cardinal', ' كرت', ' ไพ่']

◆ Top-Level Feature 1614 (Activation: 0.0313)
  Words that maximally activate this feature:
  ['极为', '变了', '特别的', '但这', '门外']
    ↳ Low-Level Feature: 19 (Activation: 0.00005417)
      Words that maximally activate this low-level feature:
      ['酒', '这不', '可以说是', '不论', '紧接着']
```

(d) No Orthogonality or L1

Figure 7: Ablation studies on the unembedding matrix show no obvious advantage from the orthogonality or $\ell_1$ regularizers, though we chose to keep them for our final runs on the embeddings.

## B.3 Importance of The Causal Inner Product

As briefly mentioned in our pre-processing section, there are theoretical reasons to believe that the Euclidean inner product does not align with the underlying geometry of the unembedding space. Our results below (in Figures 14, 15, 16, 17, 18, 19, and 20) validate this hypothesis, as we find that the unwhitened unembedding matrix does not yield meaningful features. The features are completely different from the baseline, and do not seem to have any coherent meaning, as illustrated below.

```
Explanations for word: 'Queen'
================================================

◆ Top-Level Feature 951 (Activation: 0.5322)
  Words that maximally activate this feature:
  [' King', ' king', 'King', ' kings', ' KING']
  ↳ Low-Level Feature: 24 (Activation: -0.00093935)
    Words that maximally activate this low-level feature:
    [' crowns', 'könig', ' crowns', ' crown', ' roy']

◆ Top-Level Feature 1724 (Activation: 0.2876)
  Words that maximally activate this feature:
  [' lady', ' woman', ' Lady', 'lady', 'Lady']
  ↳ Low-Level Feature: 15 (Activation: 0.00006847)
    Words that maximally activate this low-level feature:
    ['girlfriend', ' girlfriend', ' Girlfriend', ' girlfriends', ' madam']

◆ Top-Level Feature 1247 (Activation: 0.2281)
  Words that maximally activate this feature:
  [' Qu', ' qu', 'Qu', ' QU', 'qu']
  ↳ Low-Level Feature: 13 (Activation: 0.00012654)
    Words that maximally activate this low-level feature:
    ['QUEUE', ' Que', 'Que', ' QUE', 'QUE']

◆ Top-Level Feature 1914 (Activation: 0.1337)
  Words that maximally activate this feature:
  [' Officer', ' Engineer', ' Professor', ' Trainer', ' Surgeon']
  ↳ Low-Level Feature: 29 (Activation: 0.00004217)
    Words that maximally activate this low-level feature:
    [' Presidents', ' Blogger', ' 4', ' Maestro', ' Appellant']

◆ Top-Level Feature 939 (Activation: 0.0612)
  Words that maximally activate this feature:
  [' chairman', ' Chairman', 'chairman', 'Chairman', ' CEO']
  ↳ Low-Level Feature: 8 (Activation: 0.00004819)
    Words that maximally activate this low-level feature:
    [' President', ' president', 'President', 'president', ' PRESIDENT']
```

(a) Baseline

```
Explanations for word: 'Queen'
================================================

◆ Top-Level Feature 712 (Activation: 0.5264)
  Words that maximally activate this feature:
  [' King', ' king', 'King', ' kings', ' KING']
  ↳ Low-Level Feature: 12 (Activation: -0.00047339)
    Words that maximally activate this low-level feature:
    [' Royal', ' royal', 'Royal', 'royal', ' ROYAL']

◆ Top-Level Feature 1247 (Activation: 0.2378)
  Words that maximally activate this feature:
  [' qu', ' Qu', 'Qu', ' QU', 'QU']
  ↳ Low-Level Feature: 23 (Activation: 0.00012804)
    Words that maximally activate this low-level feature:
    [' الق', 'الق' , ' Que', 'QUE', 'ques']

◆ Top-Level Feature 1051 (Activation: 0.1930)
  Words that maximally activate this feature:
  [' girl', ' girls', ' Girl', 'girl', ' female']
  ↳ Low-Level Feature: 17 (Activation: 0.00011276)
    Words that maximally activate this low-level feature:
    [' Lady', 'Lady', ' lady', 'lady', ' LADY']

◆ Top-Level Feature 1911 (Activation: 0.1788)
  Words that maximally activate this feature:
  [' mama', ' Mama', 'Mama', ' mamma', 'mama']
  ↳ Low-Level Feature: 29 (Activation: 0.12758416)
    Words that maximally activate this low-level feature:
    ['madre', ' Baba', 'e', ' mère', 'mother']

◆ Top-Level Feature 1260 (Activation: 0.0884)
  Words that maximally activate this feature:
  [' Father', ' Leader', ' Contractor', ' Supervisor', ' Resident']
  ↳ Low-Level Feature: 26 (Activation: 0.00015458)
    Words that maximally activate this low-level feature:
    [' Ambassador', 'Ambassador', ' Treasurer', ' Lieutenant', ' Auditor']
```

(b) No Orthogonality

```
Explanations for word: 'Queen'
================================================

◆ Top-Level Feature 225 (Activation: 0.5392)
  Words that maximally activate this feature:
  [' king', ' King', 'King', ' kings', ' KING']
  ↳ Low-Level Feature: 29 (Activation: 0.00001101)
    Words that maximally activate this low-level feature:
    [' Princesa', '公主', ' wang', ' Reino', ' Regno']

◆ Top-Level Feature 275 (Activation: 0.2505)
  Words that maximally activate this feature:
  [' Qu', ' qu', 'Qu', ' QU', ' qu']
  ↳ Low-Level Feature: 20 (Activation: 0.00013162)
    Words that maximally activate this low-level feature:
    ['quần', ' Q', ' q', ' Qs', 'q']

◆ Top-Level Feature 863 (Activation: 0.1929)
  Words that maximally activate this feature:
  [' women', ' woman', ' Women', ' Woman', 'women']
  ↳ Low-Level Feature: 17 (Activation: 0.00006456)
    Words that maximally activate this low-level feature:
    [' dames', 'princess', 'femme', ' femmes', 'ladies']

◆ Top-Level Feature 1988 (Activation: 0.1217)
  Words that maximally activate this feature:
  [' Coach', ' Officer', ' Seller', ' Supervisor', ' Engineer']
  ↳ Low-Level Feature: 19 (Activation: 0.04881151)
    Words that maximally activate this low-level feature:
    [' Engineer', 'Engineer', ' Philosopher', 'Executive', ' Mama']

◆ Top-Level Feature 1560 (Activation: 0.0386)
  Words that maximally activate this feature:
  [' sheet', 'sheet', ' Sheet', 'Sheet', ' sheets']
  ↳ Low-Level Feature: 18 (Activation: 0.00012621)
    Words that maximally activate this low-level feature:
    ['worksheet', 'шш', '체', ' feuille', 'Workbook']
```

(c) No L1

```
Explanations for word: 'Queen'
================================================

◆ Top-Level Feature 1253 (Activation: 0.5076)
  Words that maximally activate this feature:
  [' King', ' king', 'King', ' kings', ' KING']
  ↳ Low-Level Feature: 3 (Activation: 0.00013319)
    Words that maximally activate this low-level feature:
    ['crown', ' принце', '왕', ' hoàng', '太子']

◆ Top-Level Feature 1428 (Activation: 0.3291)
  Words that maximally activate this feature:
  [' woman', ' lady', ' Woman', ' women', 'woman']
  ↳ Low-Level Feature: 13 (Activation: 0.00006558)
    Words that maximally activate this low-level feature:
    [' nuns', ' 少女', 'LADY', ' actresses', ' heiress']

◆ Top-Level Feature 1849 (Activation: 0.2592)
  Words that maximally activate this feature:
  [' Q', ' q', 'Q', 'q', ' qu']
  ↳ Low-Level Feature: 5 (Activation: -0.00062655)
    Words that maximally activate this low-level feature:
    [' Quebec', 'Quebec', 'QUAD', ' Quadrant', ' Quests']

◆ Top-Level Feature 1967 (Activation: 0.1479)
  Words that maximally activate this feature:
  [' Governor', ' Father', ' Supervisor', ' Lieutenant', ' Secretary']
  ↳ Low-Level Feature: 8 (Activation: 0.00014508)
    Words that maximally activate this low-level feature:
    [' Elder', 'Elder', ' Governor', ' Inventor', ' Philosopher']

◆ Top-Level Feature 315 (Activation: 0.0570)
  Words that maximally activate this feature:
  [' queue', ' queues', ' Queue', 'queue', 'Queue']
  ↳ Low-Level Feature: 10 (Activation: -0.00282054)
    Words that maximally activate this low-level feature:
    [' enqueue', ' Que', '큐', ' enqueue', ' оче']
```

(d) No Orthogonality or L1

Figure 8: Ablation studies on the unembedding matrix show no obvious advantage from the orthogonality or $\ell_1$ regularizers, though we chose to keep them for our final runs on the embeddings.

```
Explanations for word: 'Chicago'
================================================

◆ Top-Level Feature 1927 (Activation: 0.4056)
  Words that maximally activate this feature:
  [' Toronto', ' Chicago', ' Atlanta', ' Mumbai', ' Denver']
  ↳ Low-Level Feature: 10 (Activation: 0.00009848)
    Words that maximally activate this low-level feature:
    [' Madrid', 'Madrid', ' Barcelona', 'Barcelona', ' Tucson']

◆ Top-Level Feature 859 (Activation: 0.1282)
  Words that maximally activate this feature:
  [' American', ' America', ' american', 'American', ' Americans']
  ↳ Low-Level Feature: 5 (Activation: 0.00002996)
    Words that maximally activate this low-level feature:
    [' USA', 'USA', ' أمريكا', ' EUA', ' США']

◆ Top-Level Feature 175 (Activation: 0.1170)
  Words that maximally activate this feature:
  [' Texas', ' Alabama', ' Florida', ' Louisiana', 'Texas']
  ↳ Low-Level Feature: 19 (Activation: 0.05223560)
    Words that maximally activate this low-level feature:
    ['iowa', ' Iowa', ' Kansas', 'Iowa', 'Kansas']

◆ Top-Level Feature 98 (Activation: 0.1094)
  Words that maximally activate this feature:
  [' Chrom', ' chrom', 'chrom', ' Chrome', 'Chrom']
  ↳ Low-Level Feature: 0 (Activation: -0.00002403)
    Words that maximally activate this low-level feature:
    [' Chromosome', 'Chromosome', ' browser', ' thermo', ' nhiễm']

◆ Top-Level Feature 1512 (Activation: 0.0887)
  Words that maximally activate this feature:
  [' il', 'il', ' IL', ' IL', 'Il']
  ↳ Low-Level Feature: 8 (Activation: 0.00015619)
    Words that maximally activate this low-level feature:
    [' ila', ' Illusion', ' ilo', 'ilation', 'ил']
```

(a) Baseline

```
Explanations for word: 'Chicago'
================================================

◆ Top-Level Feature 1622 (Activation: 0.2890)
  Words that maximally activate this feature:
  [' Paris', ' Berlin', ' Tokyo', ' Madrid', ' Nairobi']
  ↳ Low-Level Feature: 4 (Activation: 0.05622706)
    Words that maximally activate this low-level feature:
    [' Chicago', 'Chicago', ' CHICAGO', ' chicago', ' Boston']

◆ Top-Level Feature 805 (Activation: 0.1590)
  Words that maximally activate this feature:
  [' Illinois', ' Missouri', ' Wisconsin', ' Michigan', ' Minnesota']
  ↳ Low-Level Feature: 13 (Activation: 0.00003956)
    Words that maximally activate this low-level feature:
    ['Delaware', ' Delaware', ' Montana', 'Dakota', ' Dakota']

◆ Top-Level Feature 546 (Activation: 0.0862)
  Words that maximally activate this feature:
  [' Ch', ' ch', ' CH', 'Ch', ' Cho']
  ↳ Low-Level Feature: 11 (Activation: 0.00009995)
    Words that maximally activate this low-level feature:
    [' CHI', 'echa', 'chir', ' chir', 'chim']

◆ Top-Level Feature 1756 (Activation: 0.0847)
  Words that maximally activate this feature:
  [' il', ' Il', ' Ill', 'il', 'Il']
  ↳ Low-Level Feature: 14 (Activation: -0.00006478)
    Words that maximally activate this low-level feature:
    ['ᐃᓄ', 'ILT', 'cile', 'sill', 'cil']

◆ Top-Level Feature 175 (Activation: 0.0654)
  Words that maximally activate this feature:
  [' American', 'American', ' american', ' Americans', ' America']
  ↳ Low-Level Feature: 11 (Activation: 0.00004174)
    Words that maximally activate this low-level feature:
    [' Амери', ' Amerikan', ' США', '美国的', ' AMERICAN']
```

(b) No Orthogonality

```
Explanations for word: 'Chicago'
================================================

◆ Top-Level Feature 904 (Activation: 0.2868)
  Words that maximally activate this feature:
  [' Tokyo', ' Berlin', ' Madrid', ' Delhi', ' London']
  ↳ Low-Level Feature: 5 (Activation: 0.07947742)
    Words that maximally activate this low-level feature:
    [' Yogyakarta', ' Boston', 'Boston', 'boston', ' CHICAGO']

◆ Top-Level Feature 937 (Activation: 0.1280)
  Words that maximally activate this feature:
  [' Chrom', ' chrom', ' Chrome', 'chrom', 'Chrom']
  ↳ Low-Level Feature: 2 (Activation: -0.00002368)
    Words that maximally activate this low-level feature:
    [' browser', ' Browser', ' browsers', 'browser', 'Browser']

◆ Top-Level Feature 1815 (Activation: 0.0733)
  Words that maximally activate this feature:
  [' California', ' Pennsylvania', ' Massachusetts', 'California', ' Connecticut']
  ↳ Low-Level Feature: 24 (Activation: 0.03427194)
    Words that maximally activate this low-level feature:
    [' Iowa', 'Iowa', ' Missouri', ' Nebraska', ' Wisconsin']

◆ Top-Level Feature 1908 (Activation: 0.0595)
  Words that maximally activate this feature:
  [' Che', ' CHE', 'che', ' che', 'Che']
  ↳ Low-Level Feature: 16 (Activation: -0.00006747)
    Words that maximally activate this low-level feature:
    [' Chess', 'Chess', ' chess', 'chess', ' Ches']

◆ Top-Level Feature 1293 (Activation: 0.0562)
  Words that maximally activate this feature:
  [' American', ' Americans', ' America', 'American', ' american']
  ↳ Low-Level Feature: 9 (Activation: 0.00003683)
    Words that maximally activate this low-level feature:
    [' statunitense', ' الأمريكي', ' estadounidense', ' 미국', ' 美国']
```

(c) No L1

```
Explanations for word: 'Chicago'
================================================

◆ Top-Level Feature 904 (Activation: 0.2514)
  Words that maximally activate this feature:
  [' Tokyo', ' Paris', ' Madrid', ' Berlin', ' Delhi']
  ↳ Low-Level Feature: 17 (Activation: 0.05849538)
    Words that maximally activate this low-level feature:
    [' NYC', ' CHICAGO', ' Brooklyn', ' Chicago', ' Boston']

◆ Top-Level Feature 1246 (Activation: 0.0796)
  Words that maximally activate this feature:
  [' American', ' Americans', ' America', ' american', 'American']
  ↳ Low-Level Feature: 9 (Activation: 0.04920823)
    Words that maximally activate this low-level feature:
    [' américain', ' amerikan', ' amer', ' statunitense', ' USD']

◆ Top-Level Feature 624 (Activation: 0.0655)
  Words that maximally activate this feature:
  [' ch', ' Ch', 'Ch', 'ch', ' CH']
  ↳ Low-Level Feature: 18 (Activation: 0.00012482)
    Words that maximally activate this low-level feature:
    [' chi', ' Chi', 'Chi', 'chi', 'CHI']

◆ Top-Level Feature 1584 (Activation: 0.0599)
  Words that maximally activate this feature:
  [' il', ' Il', 'il', 'Il', ' Ill']
  ↳ Low-Level Feature: 20 (Activation: 0.00000981)
    Words that maximally activate this low-level feature:
    [' Wil', 'ष्ट', 'ILL', 'ildo', 'ilos']

◆ Top-Level Feature 1983 (Activation: 0.0564)
  Words that maximally activate this feature:
  [' chemical', ' Chemical', 'chemical', 'Chemical', ' chemicals']
  ↳ Low-Level Feature: 30 (Activation: 0.00002974)
    Words that maximally activate this low-level feature:
    [' química', ' kimia', ' chemists', ' 화', 'สาร']
```

(d) No Orthogonality or L1

Figure 9: Ablation studies on the unembedding matrix show no obvious advantage from the orthogonality or $\ell_1$ regularizers, though we chose to keep them for our final runs on the embeddings.

```
Explanations for word: 'London'
===============================================

◆ Top-Level Feature 1927 (Activation: 0.3059)
  Words that maximally activate this feature:
  [' Toronto', ' Chicago', ' Atlanta', ' Mumbai', ' Denver']
  ↳ Low-Level Feature: 20 (Activation: 0.05103210)
    Words that maximally activate this low-level feature:
    [' Cairo', 'Cairo', 'cairo', ' London', 'London']

◆ Top-Level Feature 1197 (Activation: 0.1395)
  Words that maximally activate this feature:
  [' English', ' english', 'English', ' ENGLISH', 'english']
  ↳ Low-Level Feature: 22 (Activation: 0.10184363)
    Words that maximally activate this low-level feature:
    ['British', ' british', 'british', ' Великобрита', '▮']

◆ Top-Level Feature 1109 (Activation: 0.0617)
  Words that maximally activate this feature:
  [' Italy', ' Spain', ' India', ' Sweden', ' Greece']
  ↳ Low-Level Feature: 8 (Activation: 0.00007217)
    Words that maximally activate this low-level feature:
    [' Brasil', ' Brasil', ' BRASIL', ' BRAZIL', 'brasil']

◆ Top-Level Feature 1494 (Activation: 0.0328)
  Words that maximally activate this feature:
  [' layer', ' Layer', ' layers', 'layer', ' Layer']
  ↳ Low-Level Feature: 5 (Activation: 0.00003101)
    Words that maximally activate this low-level feature:
    ['lay', 'LAY', 'Lay', ' Lay', ' lay']

◆ Top-Level Feature 780 (Activation: 0.0312)
  Words that maximally activate this feature:
  [' lamp', ' Lamp', ' amp', 'lamp', ' amp']
  ↳ Low-Level Feature: 14 (Activation: 0.00011100)
    Words that maximally activate this low-level feature:
    ['mps', ' amplifiers', '🔊', 'amping', 'ɔ']
```

(a) Baseline

```
Explanations for word: 'London'
===============================================

◆ Top-Level Feature 1246 (Activation: 0.4808)
  Words that maximally activate this feature:
  [' UK', 'UK', ' London', 'London', ' Britain']
  ↳ Low-Level Feature: 12 (Activation: -0.00007963)
    Words that maximally activate this low-level feature:
    [' Gloucestershire', ' Warwickshire', ' Hertfordshire', ' Wiltshire', ' Oxfordshire']

◆ Top-Level Feature 1622 (Activation: 0.4351)
  Words that maximally activate this feature:
  [' Paris', ' Berlin', ' Tokyo', ' Madrid', ' Nairobi']
  ↳ Low-Level Feature: 26 (Activation: 0.00009957)
    Words that maximally activate this low-level feature:
    [' Lagos', 'Lagos', ' Abuja', ' Brussels', ' Abuja']

◆ Top-Level Feature 1915 (Activation: 0.0706)
  Words that maximally activate this feature:
  [' L', 'L', ' Л', ' LC', ' LR']
  ↳ Low-Level Feature: 20 (Activation: 0.00004870)
    Words that maximally activate this low-level feature:
    [' LOR', 'LOR', ' Loren', ' Lorde', 'LLE']

◆ Top-Level Feature 1109 (Activation: 0.0691)
  Words that maximally activate this feature:
  [' Spain', ' Italy', ' Poland', ' Russia', ' Sweden']
  ↳ Low-Level Feature: 20 (Activation: 0.00008624)
    Words that maximally activate this low-level feature:
    [' Angola', 'ジャパン', ' Sweden', ' Azerbaijan', ' Algeria']

◆ Top-Level Feature 1197 (Activation: 0.0492)
  Words that maximally activate this feature:
  [' English', ' english', 'English', ' ENGLISH', 'english']
  ↳ Low-Level Feature: 15 (Activation: 0.00001979)
    Words that maximally activate this low-level feature:
    [' anglo', '英语', ' inglese', ' Engl', ' Анг']
```

(b) No Orthogonality

```
Explanations for word: 'London'
===============================================

◆ Top-Level Feature 904 (Activation: 0.4297)
  Words that maximally activate this feature:
  [' Tokyo', ' Berlin', ' Madrid', ' Delhi', ' London']
  ↳ Low-Level Feature: 27 (Activation: -0.00001157)
    Words that maximally activate this low-level feature:
    [' Helsinki', 'Helsinki', ' Copenhagen', ' Tallinn', ' Киев']

◆ Top-Level Feature 1796 (Activation: 0.1850)
  Words that maximally activate this feature:
  [' Nottingham', ' Leicester', ' Lancashire', ' Yorkshire', ' Gloucestershire']
  ↳ Low-Level Feature: 29 (Activation: 0.06130284)
    Words that maximally activate this low-level feature:
    [' UK', 'UK', ' britannique', 'イギリス', ' britann']

◆ Top-Level Feature 1233 (Activation: 0.0799)
  Words that maximally activate this feature:
  [' L', 'L', ' Л', 'Л', ' LR']
  ↳ Low-Level Feature: 12 (Activation: 0.00002894)
    Words that maximally activate this low-level feature:
    [' Lu', ' lu', ' LU', ' Luc', 'LU']

◆ Top-Level Feature 1174 (Activation: 0.0453)
  Words that maximally activate this feature:
  [' Spain', ' Canada', ' Italy', ' France', ' Germany']
  ↳ Low-Level Feature: 24 (Activation: -0.00002373)
    Words that maximally activate this low-level feature:
    [' Italy', ' Italia', 'Italy', ' italy', ' ITALY']

◆ Top-Level Feature 1815 (Activation: 0.0342)
  Words that maximally activate this feature:
  [' California', ' Pennsylvania', ' Massachusetts', 'California', ' Connecticut']
  ↳ Low-Level Feature: 12 (Activation: 0.00007537)
    Words that maximally activate this low-level feature:
    [' Sonoma', ' Alabama', 'Alabama', ' California', ' Texas']
```

(c) No L1

```
Explanations for word: 'London'
===============================================

◆ Top-Level Feature 904 (Activation: 0.3255)
  Words that maximally activate this feature:
  [' Tokyo', ' Paris', ' Madrid', ' Berlin', ' Delhi']
  ↳ Low-Level Feature: 26 (Activation: 0.08774494)
    Words that maximally activate this low-level feature:
    [' London', '伦敦', ' dubai', 'PARIS', 'ローマ']

◆ Top-Level Feature 1254 (Activation: 0.3215)
  Words that maximally activate this feature:
  [' British', ' Britain', 'British', ' UK', ' Brits']
  ↳ Low-Level Feature: 25 (Activation: 0.00004446)
    Words that maximally activate this low-level feature:
    [' Scotsman', ' 영', '▮', ' Britton', ' Brexit']

◆ Top-Level Feature 1873 (Activation: 0.0714)
  Words that maximally activate this feature:
  [' Italy', ' Ireland', ' France', ' Brazil', ' Poland']
  ↳ Low-Level Feature: 10 (Activation: 0.00006569)
    Words that maximally activate this low-level feature:
    ['indonesia', ' Jamaica', ' INDONESIA', 'ישראל', ' Polsce']

◆ Top-Level Feature 1425 (Activation: 0.0353)
  Words that maximally activate this feature:
  [' L', 'L', ' Л', ' LC', ' LR']
  ↳ Low-Level Feature: 19 (Activation: 0.00001737)
    Words that maximally activate this low-level feature:
    [' LUIS', ' LSM', ' 劉', '劉', 'Ľ']

◆ Top-Level Feature 1109 (Activation: 0.0353)
  Words that maximally activate this feature:
  [' the', ' charge', ' flow', ' not', ' schedule']
  ↳ Low-Level Feature: 2 (Activation: 0.00006976)
    Words that maximally activate this low-level feature:
    [' the', ' not', ' email', ' sheet', ' the']
```

(d) No Orthogonality or L1

Figure 10: Ablation studies on the unembedding matrix show no obvious advantage from the orthogonality or $\ell_1$ regularizers, though we chose to keep them for our final runs on the embeddings.

```
Explanations for word: 'Twitter'
============================================

◆ Top-Level Feature 1959 (Activation: 0.5850)
  Words that maximally activate this feature:
  [' tweet', ' tweets', ' Tweet', ' tweet', ' tweeting']
  ↳ Low-Level Feature: 24 (Activation: 0.00011509)
      Words that maximally activate this low-level feature:
      [' tweeting', ' twitter', ' Twitter', ' TWITTER', 'twitter']

◆ Top-Level Feature 1939 (Activation: 0.3562)
  Words that maximally activate this feature:
  [' YouTube', ' Youtube', ' youtube', ' Google', 'YouTube']
  ↳ Low-Level Feature: 7 (Activation: -0.00020170)
      Words that maximally activate this low-level feature:
      [' goog', ' Flickr', 'yahoo', 'Wikipedia', 'Wiki']

◆ Top-Level Feature 506 (Activation: 0.1397)
  Words that maximally activate this feature:
  [' social', ' Social', 'Social', 'social', ' SOCIAL']
  ↳ Low-Level Feature: 3 (Activation: -0.00001523)
      Words that maximally activate this low-level feature:
      [' socialists', ' socialist', ' Socialist', ' Socialists', ' socialism']

◆ Top-Level Feature 1136 (Activation: 0.0638)
  Words that maximally activate this feature:
  [' blockchain', ' Blockchain', ' NFTs', ' TikTok', 'Blockchain']
  ↳ Low-Level Feature: 27 (Activation: 0.00007173)
      Words that maximally activate this low-level feature:
      ['新冠', ' Trump', 'Trump', ' Biden', ' https']

◆ Top-Level Feature 2046 (Activation: 0.0511)
  Words that maximally activate this feature:
  [' Tuesday', ' Wednesday', ' Thursday', ' Monday', ' Friday']
  ↳ Low-Level Feature: 22 (Activation: 0.00005836)
      Words that maximally activate this low-level feature:
      [' Sunday', 'Sunday', ' sunday', ' SUNDAY', 'SUNDAY']
```

(a) Baseline

```
Explanations for word: 'Twitter'
============================================

◆ Top-Level Feature 1694 (Activation: 0.4420)
  Words that maximally activate this feature:
  [' Facebook', ' facebook', ' YouTube', 'Facebook', ' Youtube']
  ↳ Low-Level Feature: 31 (Activation: 0.11588523)
      Words that maximally activate this low-level feature:
      [' tweet', ' Tweet', 'Tweet', ' twitter', ' Twitter']

◆ Top-Level Feature 902 (Activation: 0.0544)
  Words that maximally activate this feature:
  [' social', ' Social', 'social', 'Social', ' SOCIAL']
  ↳ Low-Level Feature: 16 (Activation: -0.01861620)
      Words that maximally activate this low-level feature:
      [' SOC', 'soc', '社会', ' sociais', ' soc']

◆ Top-Level Feature 1939 (Activation: 0.0433)
  Words that maximally activate this feature:
  [' web', ' Web', 'web', 'Web', ' WEB']
  ↳ Low-Level Feature: 4 (Activation: 0.00003201)
      Words that maximally activate this low-level feature:
      ['ウェブ', ' weber', ' webview', ' webs', ' webpage']

◆ Top-Level Feature 1210 (Activation: 0.0392)
  Words that maximally activate this feature:
  [' Java', 'Java', ' JAVA', ' java', ' PHP']
  ↳ Low-Level Feature: 22 (Activation: 0.00003138)
      Words that maximally activate this low-level feature:
      ['MySQL', 'Mysql', 'mysql', ' MySQL', ' Kubernetes']

◆ Top-Level Feature 1549 (Activation: 0.0377)
  Words that maximally activate this feature:
  [' Smartphone', ' smartphone', ' Smartphones', ' smartphones', 'smartphone']
  ↳ Low-Level Feature: 5 (Activation: 0.00001795)
      Words that maximally activate this low-level feature:
      ['iPhone', ' iPhone', 'iphone', ' iphone', 'iPad']
```

(b) No Orthogonality

```
Explanations for word: 'Twitter'
============================================

◆ Top-Level Feature 512 (Activation: 0.5074)
  Words that maximally activate this feature:
  [' tweet', ' tweets', ' Tweet', 'tweet', ' tweeting']
  ↳ Low-Level Feature: 26 (Activation: 0.00000981)
      Words that maximally activate this low-level feature:
      [' Twit', 'Twe', ' Hashtag', 'twe', ' Twitter']

◆ Top-Level Feature 1403 (Activation: 0.3461)
  Words that maximally activate this feature:
  [' YouTube', ' Youtube', ' Pinterest', ' LinkedIn', 'YouTube']
  ↳ Low-Level Feature: 30 (Activation: 0.04850825)
      Words that maximally activate this low-level feature:
      ['instagram', ' instagram', ' INSTAGRAM', 'Instagram', ' Instagram']

◆ Top-Level Feature 1549 (Activation: 0.0641)
  Words that maximally activate this feature:
  [' social', ' Social', 'social', 'Social', ' SOCIAL']
  ↳ Low-Level Feature: 14 (Activation: -0.00001868)
      Words that maximally activate this low-level feature:
      [' sociala', 'Social', ' socially', ' Social', ' sosial']

◆ Top-Level Feature 450 (Activation: 0.0560)
  Words that maximally activate this feature:
  [' blog', ' Blog', ' blogs', 'Blog', 'blog']
  ↳ Low-Level Feature: 0 (Activation: -0.00017220)
      Words that maximally activate this low-level feature:
      [' article', ' artikel', 'Artikel', '博客', ' Blog']

◆ Top-Level Feature 1563 (Activation: 0.0417)
  Words that maximally activate this feature:
  [' web', ' Web', 'web', 'Web', ' WEB']
  ↳ Low-Level Feature: 26 (Activation: 0.00013382)
      Words that maximally activate this low-level feature:
      [' Internet', 'Internet', 'internet', ' internet', ' INTERNET']
```

(c) No L1

```
Explanations for word: 'Twitter'
============================================

◆ Top-Level Feature 1618 (Activation: 0.6319)
  Words that maximally activate this feature:
  [' tweet', ' tweets', ' twitter', ' Tweet', ' Twitter']
  ↳ Low-Level Feature: 17 (Activation: 0.00008684)
      Words that maximally activate this low-level feature:
      [' Blogging', ' blogging', 'Blogger', ' TWITTER', ' hashtag']

◆ Top-Level Feature 1045 (Activation: 0.1111)
  Words that maximally activate this feature:
  [' social', ' Social', 'Social', 'social', ' SOCIAL']
  ↳ Low-Level Feature: 1 (Activation: 0.02887224)
      Words that maximally activate this low-level feature:
      [' Facebook', 'Facebook', 'facebook', 'FACEBOOK', ' sociable']

◆ Top-Level Feature 1245 (Activation: 0.0928)
  Words that maximally activate this feature:
  [' Disney', ' Hasbro', ' Pfizer', ' Nestlé', 'Disney']
  ↳ Low-Level Feature: 16 (Activation: 0.00003986)
      Words that maximally activate this low-level feature:
      [' FAO', ' Walgreens', ' Cisco', ' Daimler', ' UNHCR']

◆ Top-Level Feature 578 (Activation: 0.0844)
  Words that maximally activate this feature:
  [' blockchain', ' Blockchain', ' TikTok', ' cryptocurrency', ' NFTs']
  ↳ Low-Level Feature: 12 (Activation: -0.00022477)
      Words that maximally activate this low-level feature:
      ['👉', 'CRYPTO', ' Crypto', 'crypto', ' Vegan']

◆ Top-Level Feature 1594 (Activation: 0.0561)
  Words that maximally activate this feature:
  [' telephone', ' Telephone', 'telephone', ' phone', 'Telephone']
  ↳ Low-Level Feature: 25 (Activation: -0.00012464)
      Words that maximally activate this low-level feature:
      [' Sms', 'iphone', ' telemetry', ' télé', 'Tele']
```

(d) No Orthogonality or L1

Figure 11: Ablation studies on the unembedding matrix show no obvious advantage from the orthogonality or $\ell_1$ regularizers, though we chose to keep them for our final runs on the embeddings.

```
Explanations for word: 'python'
================================================
◆ Top-Level Feature 547 (Activation: 0.0955)
  Words that maximally activate this feature:
  [' pit', ' Pit', 'Pit', 'pit', ' pits']
  ↳ Low-Level Feature: 9 (Activation: 0.07605679)
      Words that maximally activate this low-level feature:
      [' py', 'Python', ' Python', ' python', 'python']

◆ Top-Level Feature 734 (Activation: 0.0839)
  Words that maximally activate this feature:
  [' script', ' Script', ' scripts', 'Script', 'script']
  ↳ Low-Level Feature: 4 (Activation: 0.00004199)
      Words that maximally activate this low-level feature:
      ['Script']

◆ Top-Level Feature 51 (Activation: 0.0826)
  Words that maximally activate this feature:
  [' snap', ' snaps', ' Sna', 'sna', 'snap']
  ↳ Low-Level Feature: 16 (Activation: -0.00001486)
      Words that maximally activate this low-level feature:
      ['Snap', ' snapping', 'SNA', ' Sni', ' sna']

◆ Top-Level Feature 1867 (Activation: 0.0624)
  Words that maximally activate this feature:
  [' Checkbox', ' DataBase', ' TextBox', ' mongodb', ' ToDo']
  ↳ Low-Level Feature: 19 (Activation: 0.06658750)
      Words that maximally activate this low-level feature:
      [' Javascript', ' javascript', ' JAVA', ' Kotlin', ' TypeScript']

◆ Top-Level Feature 1316 (Activation: 0.0506)
  Words that maximally activate this feature:
  [' ph', ' Ph', 'ph', 'Ph', ' PH']
  ↳ Low-Level Feature: 17 (Activation: 0.00003440)
      Words that maximally activate this low-level feature:
      [' Philip', ' Phillip', 'Philip', ' Philips', 'Phillip']
```

(a) Baseline

```
Explanations for word: 'python'
================================================
◆ Top-Level Feature 770 (Activation: 0.4408)
  Words that maximally activate this feature:
  [' Py', ' py', 'Py', 'py', ' PY']
  ↳ Low-Level Feature: 25 (Activation: -0.00062331)
      Words that maximally activate this low-level feature:
      ['numpy', 'pandas', ' numpy', 'django', 'pygame']

◆ Top-Level Feature 1210 (Activation: 0.3428)
  Words that maximally activate this feature:
  [' Java', 'Java', ' JAVA', ' java', ' PHP']
  ↳ Low-Level Feature: 22 (Activation: 0.00007985)
      Words that maximally activate this low-level feature:
      ['MySQL', 'Mysql', 'mysql', ' MySQL', ' Kubernetes']

◆ Top-Level Feature 1293 (Activation: 0.0737)
  Words that maximally activate this feature:
  [' tiger', ' Tiger', 'Tiger', ' tigers', ' elephant']
  ↳ Low-Level Feature: 21 (Activation: 0.00015797)
      Words that maximally activate this low-level feature:
      [' Dinosaur', ' alligator', ' Turtle', ' Dragons', ' turtle']

◆ Top-Level Feature 1950 (Activation: 0.0728)
  Words that maximally activate this feature:
  [' script', ' Script', ' scripts', 'Script', 'script']
  ↳ Low-Level Feature: 5 (Activation: -0.00019795)
      Words that maximally activate this low-level feature:
      [' scripted', ' SCRIPT', ' Scrip', 'SCRIPT', ' сцена']

◆ Top-Level Feature 1246 (Activation: 0.0440)
  Words that maximally activate this feature:
  [' UK', 'UK', ' London', 'London', ' Britain']
  ↳ Low-Level Feature: 19 (Activation: 0.00006086)
      Words that maximally activate this low-level feature:
      [' england', ' london', ' british', 'ukh', ' Hertfordshire']
```

(b) No Orthogonality

```
Explanations for word: 'python'
================================================
◆ Top-Level Feature 1294 (Activation: 0.4444)
  Words that maximally activate this feature:
  [' Java', ' JAVA', 'Java', ' Python', ' python']
  ↳ Low-Level Feature: 5 (Activation: -0.00015276)
      Words that maximally activate this low-level feature:
      [' Postgres', ' postgres', ' PostgreSQL', ' MySQL', 'MySQL']

◆ Top-Level Feature 2039 (Activation: 0.1708)
  Words that maximally activate this feature:
  [' snake', ' Snake', 'snake', 'Snake', ' snakes']
  ↳ Low-Level Feature: 29 (Activation: 0.00013670)
      Words that maximally activate this low-level feature:
      [' viper', 'viper', ' venomous', ' cobra', ' scorpion']

◆ Top-Level Feature 823 (Activation: 0.1585)
  Words that maximally activate this feature:
  [' pi', ' Pi', 'pi', 'Pi', ' PI']
  ↳ Low-Level Feature: 16 (Activation: 0.00014878)
      Words that maximally activate this low-level feature:
      [' piety', ' PIC', ' pi', 'Пи', ' Pyramid']

◆ Top-Level Feature 1622 (Activation: 0.1320)
  Words that maximally activate this feature:
  ['py', ' PY', ' PY', ' Py', ' Ry']
  ↳ Low-Level Feature: 23 (Activation: -0.00012211)
      Words that maximally activate this low-level feature:
      [' Blythe', 'aly', 'ovy', 'ály', ' Nya']

◆ Top-Level Feature 1774 (Activation: 0.0688)
  Words that maximally activate this feature:
  [' pd', ' dm', ' dt', ' md', ' ml']
  ↳ Low-Level Feature: 29 (Activation: 0.00010894)
      Words that maximally activate this low-level feature:
      [' ctx', ' dll', ' dst', ' iv', ' mb']
```

(c) No L1

```
Explanations for word: 'python'
================================================
◆ Top-Level Feature 251 (Activation: 0.0905)
  Words that maximally activate this feature:
  [' hd', ' fb', ' cb', ' gps', ' pc']
  ↳ Low-Level Feature: 9 (Activation: 0.00009679)
      Words that maximally activate this low-level feature:
      [' pc', ' ai', ' ford', ' ajax', ' unicode']

◆ Top-Level Feature 798 (Activation: 0.0771)
  Words that maximally activate this feature:
  [' pi', ' Pi', 'pi', 'Pi', ' PI']
  ↳ Low-Level Feature: 26 (Activation: 0.23805813)
      Words that maximally activate this low-level feature:
      [' Python', ' piercing', 'pim', 'opi', ' Piers']

◆ Top-Level Feature 1866 (Activation: 0.0351)
  Words that maximally activate this feature:
  [' tiger', ' Tiger', ' lion', ' Lion', ' wolf']
  ↳ Low-Level Feature: 29 (Activation: 0.00000708)
      Words that maximally activate this low-level feature:
      [' turtle', ' penguin', ' Gecko', ' dolphin', ' hedgehog']

◆ Top-Level Feature 594 (Activation: 0.0280)
  Words that maximally activate this feature:
  [' script', ' Script', 'Script', 'script']
  ↳ Low-Level Feature: 27 (Activation: 0.00010754)
      Words that maximally activate this low-level feature:
      ['javascript', ' scripture', ' manuscrit', ' kịch', ' скри']

◆ Top-Level Feature 1499 (Activation: 0.0255)
  Words that maximally activate this feature:
  [' plastic', ' Plastic', 'plastic', ' plastics', ' Plas']
  ↳ Low-Level Feature: 25 (Activation: 0.01944395)
      Words that maximally activate this low-level feature:
      ['YAML', ' YAML', 'jackson', 'ایل', 'JSON']
```

(d) No Orthogonality or L1

Figure 12: Ablation studies on the unembedding matrix show no obvious advantage from the orthogonality or $\ell_1$ regularizers, though we chose to keep them for our final runs on the embeddings.

```
Explanations for word: 'Bayesian'
=========================================

◆ Top-Level Feature 901 (Activation: 0.1695)
  Words that maximally activate this feature:
  [' Jacobian', ' Dirichlet', ' bilinear', ' piecewise', ' variational']
  ↳ Low-Level Feature: 31 (Activation: 0.00010277)
     Words that maximally activate this low-level feature:
     [' singularities', ' oscillatory', ' moduli', ' sinusoidal', ' trigonometric']

◆ Top-Level Feature 2010 (Activation: 0.1214)
  Words that maximally activate this feature:
  [' probability', ' probabilities', ' Probability', 'probability', ' Probab']
  ↳ Low-Level Feature: 15 (Activation: 0.00001307)
     Words that maximally activate this low-level feature:
     [' prospects', 'Probable', ' probable', ' Probable', 'probable']

◆ Top-Level Feature 870 (Activation: 0.0941)
  Words that maximally activate this feature:
  [' Catholic', ' Hindu', 'Catholic', ' Muslim', ' Christian']
  ↳ Low-Level Feature: 31 (Activation: -0.00001035)
     Words that maximally activate this low-level feature:
     [' Aryan', ' Huguen', ' Aryan', 'Gothic', ' Gregorian']

◆ Top-Level Feature 1837 (Activation: 0.0801)
  Words that maximally activate this feature:
  [' ba', ' Ba', 'Ba', 'ba', ' BA']
  ↳ Low-Level Feature: 30 (Activation: 0.07795604)
     Words that maximally activate this low-level feature:
     [' Bá', ' bay', ' Bay', 'Bay', ' BAY']

◆ Top-Level Feature 404 (Activation: 0.0781)
  Words that maximally activate this feature:
  [' statistics', ' Statistics', ' statistical', ' statistic', ' stats']
  ↳ Low-Level Feature: 12 (Activation: 0.00004004)
     Words that maximally activate this low-level feature:
     [' thống', ' stat', ' Estad', ' statistics', ' Statistical']
```

(a) Baseline

```
Explanations for word: 'Bayesian'
=========================================

◆ Top-Level Feature 1054 (Activation: 0.0860)
  Words that maximally activate this feature:
  [' Ba', ' ba', 'Ba', 'ba', ' BA']
  ↳ Low-Level Feature: 4 (Activation: 0.11283194)
     Words that maximally activate this low-level feature:
     [' BAY', ' bays', ' Bays', ' Bayesian', ' baie']

◆ Top-Level Feature 1852 (Activation: 0.0789)
  Words that maximally activate this feature:
  [' mathematical', ' Mathematical', ' mathematics', ' math', 'Mathematical']
  ↳ Low-Level Feature: 13 (Activation: -0.00007937)
     Words that maximally activate this low-level feature:
     [' maths', ' Mathematics', ' Calculus', 'maths', 'Maths']

◆ Top-Level Feature 1934 (Activation: 0.0594)
  Words that maximally activate this feature:
  [' Einstein', ' Descartes', ' Spinoza', ' Mozart', 'Einstein']
  ↳ Low-Level Feature: 28 (Activation: 0.00009781)
     Words that maximally activate this low-level feature:
     [' Shakespeare', 'Shakespeare', ' Macbeth', ' Othello', 'Napoleon']

◆ Top-Level Feature 1210 (Activation: 0.0559)
  Words that maximally activate this feature:
  [' Java', 'Java', ' JAVA', ' java', ' PHP']
  ↳ Low-Level Feature: 22 (Activation: -0.00002095)
     Words that maximally activate this low-level feature:
     ['MySQL', 'Mysql', 'mysql', ' MySQL', ' Kubernetes']

◆ Top-Level Feature 64 (Activation: 0.0555)
  Words that maximally activate this feature:
  [' adiabatic', ' phonon', ' anisotropic', ' oscillatory', ' linearized']
  ↳ Low-Level Feature: 4 (Activation: -0.00312077)
     Words that maximally activate this low-level feature:
     [' dielectric', ' depolar', ' sintering', ' conformal', ' sinter']
```

(b) No Orthogonality

```
Explanations for word: 'Bayesian'
=========================================

◆ Top-Level Feature 1951 (Activation: 0.1547)
  Words that maximally activate this feature:
  ['ay', 'AY', ' Ay', ' ay', 'Ay']
  ↳ Low-Level Feature: 3 (Activation: 0.00001630)
     Words that maximally activate this low-level feature:
     ['zey', 'vey', ' Mey', 'aley', ' Vey']

◆ Top-Level Feature 468 (Activation: 0.1542)
  Words that maximally activate this feature:
  [' probability', ' probabilities', ' Probability', 'probability', ' Probab']
  ↳ Low-Level Feature: 30 (Activation: 0.00002666)
     Words that maximally activate this low-level feature:
     [' Probability', 'Probability', ' unlikely', ' Probab', ' probabilities']

◆ Top-Level Feature 1561 (Activation: 0.1399)
  Words that maximally activate this feature:
  [' Ba', ' ba', 'Ba', 'ba', ' BA']
  ↳ Low-Level Feature: 4 (Activation: 0.00008041)
     Words that maximally activate this low-level feature:
     [' Bail', 'Bail', ' bail', 'bail', ' bailout']

◆ Top-Level Feature 284 (Activation: 0.0992)
  Words that maximally activate this feature:
  [' anisotropic', ' adiabatic', ' oscillatory', ' phonon', ' isothermal']
  ↳ Low-Level Feature: 24 (Activation: 0.00013688)
     Words that maximally activate this low-level feature:
     [' cartesian', ' eigenvectors', ' trigonometric', ' Jacobian', ' discriminant']

◆ Top-Level Feature 1837 (Activation: 0.0812)
  Words that maximally activate this feature:
  [' statistics', ' Statistics', ' statistic', ' statistical', ' stats']
  ↳ Low-Level Feature: 24 (Activation: -0.00002423)
     Words that maximally activate this low-level feature:
     ['analytics', ' statut', 'statistic', ' stat', ' Estad']
```

(c) No L1

```
Explanations for word: 'Bayesian'
=========================================

◆ Top-Level Feature 217 (Activation: 0.1110)
  Words that maximally activate this feature:
  [' statistics', ' Statistics', ' statistic', ' statistical', 'Statistics']
  ↳ Low-Level Feature: 27 (Activation: 0.00000954)
     Words that maximally activate this low-level feature:
     ['Estat', ' statysty', ' Statisti', ' Estad', ' Statistik']

◆ Top-Level Feature 312 (Activation: 0.0925)
  Words that maximally activate this feature:
  ['AY', 'ay', 'ray', ' Ay', ' ay']
  ↳ Low-Level Feature: 14 (Activation: 0.08490946)
     Words that maximally activate this low-level feature:
     [' bay', ' Bay', 'Bay', ' BAY', 'BAY']

◆ Top-Level Feature 929 (Activation: 0.0588)
  Words that maximally activate this feature:
  [' Spinoza', ' Chaucer', ' Machiavelli', ' Proust', ' Baudelaire']
  ↳ Low-Level Feature: 21 (Activation: 0.00013820)
     Words that maximally activate this low-level feature:
     ['Newton', ' Gauss', ' Euler', 'Euler', ' Einstein']

◆ Top-Level Feature 955 (Activation: 0.0570)
  Words that maximally activate this feature:
  [' Brazilian', ' Mexican', ' Italian', ' Turkish', ' Vietnamese']
  ↳ Low-Level Feature: 5 (Activation: -0.00007113)
     Words that maximally activate this low-level feature:
     [' Italian', ' italian', ' Italians', 'Italian', 'italian']

◆ Top-Level Feature 810 (Activation: 0.0470)
  Words that maximally activate this feature:
  [' math', ' mathematics', ' Math', ' maths', 'Math']
  ↳ Low-Level Feature: 5 (Activation: 0.00011147)
     Words that maximally activate this low-level feature:
     ['algebra', ' arithmetic', ' algebraic', '数学', 'Algebra']
```

(d) No Orthogonality or L1

Figure 13: Ablation studies on the unembedding matrix show no obvious advantage from the orthogonality or $\ell_1$ regularizers, though we chose to keep them for our final runs on the embeddings.

```
Explanations for word: 'puppy'
================================================

◆ Top-Level Feature 732 (Activation: 0.3964)
  Words that maximally activate this feature:
  [' dog', ' Dog', 'Dog', 'dog', ' dogs']
    ↳ Low-Level Feature: 31 (Activation: -0.00061521)
      Words that maximally activate this low-level feature:
      [' barks', ' собаки', ' woof', ' собака', ' Labrador']

◆ Top-Level Feature 956 (Activation: 0.0883)
  Words that maximally activate this feature:
  [' pig', ' goat', ' pigs', ' monkey', ' Goat']
    ↳ Low-Level Feature: 12 (Activation: 0.00000925)
      Words that maximally activate this low-level feature:
      [' horse', 'Horse', ' Horse', ' pony', ' Pony']

◆ Top-Level Feature 117 (Activation: 0.0525)
  Words that maximally activate this feature:
  [' boy', ' Boy', 'boy', 'Boy', ' BOY']
    ↳ Low-Level Feature: 0 (Activation: 0.21417145)
      Words that maximally activate this low-level feature:
      [' menina', ' boya', 'ガール', 'BOYS', 'GIRLS']

◆ Top-Level Feature 487 (Activation: 0.0457)
  Words that maximally activate this feature:
  [' pet', ' Pet', 'pet', 'Pet', ' PET']
    ↳ Low-Level Feature: 4 (Activation: 0.00009525)
      Words that maximally activate this low-level feature:
      ['PETER', ' Pedro', 'Pedro', ' petrol', ' Petrol']

◆ Top-Level Feature 1809 (Activation: 0.0437)
  Words that maximally activate this feature:
  [' patient', ' shopper', ' attendee', ' subscriber', ' sufferer']
    ↳ Low-Level Feature: 8 (Activation: 0.00009030)
      Words that maximally activate this low-level feature:
      [' Passenger', ' passenger', 'passenger', 'Passenger', ' guest']
```

```
Explanations for word: 'puppy'
================================================

◆ Top-Level Feature 1986 (Activation: 0.1022)
  Words that maximally activate this feature:
  ['今後も', 'お花', '一歩', '本当の', 'これからも']
    ↳ Low-Level Feature: 6 (Activation: 0.04209925)
      Words that maximally activate this low-level feature:
      ['讃', '手作り', '今後も', '痛い', '一歩']

◆ Top-Level Feature 1442 (Activation: 0.1020)
  Words that maximally activate this feature:
  [' сим', ' unlicensed', 'меня', 'ohn', ' nguyên']
    ↳ Low-Level Feature: 23 (Activation: 0.06289645)
      Words that maximally activate this low-level feature:
      [' founder', ' nguyên', 'ohn', ' deflected', ' deflection']

◆ Top-Level Feature 967 (Activation: 0.0865)
  Words that maximally activate this feature:
  [' baixo', ' partidas', 'iken', 'oko', '兵']
    ↳ Low-Level Feature: 21 (Activation: 0.03897284)
      Words that maximally activate this low-level feature:
      ['Presenter', 'に限', ' договора', ' Englishman', 'то']

◆ Top-Level Feature 402 (Activation: 0.0865)
  Words that maximally activate this feature:
  ['بر', ' possess', ' RESERVE', ' Kimmel', 'Muslims']
    ↳ Low-Level Feature: 30 (Activation: 0.03146359)
      Words that maximally activate this low-level feature:
      [' upkeep', ' upkeep', ' Kimmel']

◆ Top-Level Feature 2036 (Activation: 0.0858)
  Words that maximally activate this feature:
  ['억', 'ujin', ' membunuh', ' diplom', ' im']
    ↳ Low-Level Feature: 27 (Activation: 0.02508827)
      Words that maximally activate this low-level feature:
      [' Uttarakhand', ' denial', ' heavier', ' neutrino', ' CAS']
```

(a) Baseline        (b) No Whitening

Figure 14: The causal inner product is necessary for the model to learn meaningful features.

```
Explanations for word: 'Queen'
================================================

◆ Top-Level Feature 951 (Activation: 0.5322)
  Words that maximally activate this feature:
  [' King', ' king', 'King', ' kings', ' KING']
    ↳ Low-Level Feature: 24 (Activation: -0.00093935)
      Words that maximally activate this low-level feature:
      [' crowns', 'könig', ' crowns', ' crown', ' roy']

◆ Top-Level Feature 1724 (Activation: 0.2876)
  Words that maximally activate this feature:
  [' lady', ' woman', ' Lady', 'lady', 'Lady']
    ↳ Low-Level Feature: 15 (Activation: 0.00006847)
      Words that maximally activate this low-level feature:
      ['girlfriend', ' girlfriend', ' Girlfriend', ' girlfriends', ' madam']

◆ Top-Level Feature 1247 (Activation: 0.2281)
  Words that maximally activate this feature:
  [' Qu', ' qu', 'Qu', ' QU', 'qu']
    ↳ Low-Level Feature: 13 (Activation: 0.00012654)
      Words that maximally activate this low-level feature:
      ['QUEUE', ' Que', 'Que', ' QUE', 'QUE']

◆ Top-Level Feature 1914 (Activation: 0.1337)
  Words that maximally activate this feature:
  [' Officer', ' Engineer', ' Professor', ' Trainer', ' Surgeon']
    ↳ Low-Level Feature: 29 (Activation: 0.00004217)
      Words that maximally activate this low-level feature:
      [' Presidents', ' Blogger', ' ψ', ' Maestro', ' Appellant']

◆ Top-Level Feature 939 (Activation: 0.0612)
  Words that maximally activate this feature:
  [' chairman', ' Chairman', 'chairman', 'Chairman', ' CEO']
    ↳ Low-Level Feature: 8 (Activation: 0.00004819)
      Words that maximally activate this low-level feature:
      [' President', ' president', 'President', 'president', ' PRESIDENT']
```

```
Explanations for word: 'Queen'
================================================

◆ Top-Level Feature 1707 (Activation: 0.1521)
  Words that maximally activate this feature:
  ['옹', 'M', 'v', 'ného', 'r']
    ↳ Low-Level Feature: 21 (Activation: 0.14752719)
      Words that maximally activate this low-level feature:
      [' milhões', 'かけた', 'льно', 'ToList', ' desses']

◆ Top-Level Feature 1473 (Activation: 0.1479)
  Words that maximally activate this feature:
  [' [\r', '2', '9', ' ={', '3']
    ↳ Low-Level Feature: 14 (Activation: 0.19582644)
      Words that maximally activate this low-level feature:
      [' [\r', ' ={', '4', ' [\r', '3']

◆ Top-Level Feature 1002 (Activation: 0.1417)
  Words that maximally activate this feature:
  ['МН', ' Ersten', 'meg', 'öt', 'vs']
    ↳ Low-Level Feature: 2 (Activation: 0.34217829)
      Words that maximally activate this low-level feature:
      ['МН', 'amal', 'öt', ' 劇场', 'stra']

◆ Top-Level Feature 320 (Activation: 0.1412)
  Words that maximally activate this feature:
  ['honda', ' času', 'Thời', 'esting', 'рой']
    ↳ Low-Level Feature: 0 (Activation: 0.07256843)
      Words that maximally activate this low-level feature:
      ['Array', 'Options', ' King', 'рой', ' historians']

◆ Top-Level Feature 1829 (Activation: 0.1406)
  Words that maximally activate this feature:
  ['       ', ' [', 'とりあえず', '明日', 'npy']
    ↳ Low-Level Feature: 9 (Activation: 0.05275402)
      Words that maximally activate this low-level feature:
      [' Registro', ' perpetuated', 'ـ', ' {\r', '漆']
```

(a) Baseline        (b) No Whitening

Figure 15: The causal inner product is necessary for the model to learn meaningful features.

(a) Baseline  (b) No Whitening

Figure 16: The causal inner product is necessary for the model to learn meaningful features.

(a) Baseline  (b) No Whitening

Figure 17: The causal inner product is necessary for the model to learn meaningful features.

```
Explanations for word: 'Twitter'
=================================================

◆ Top-Level Feature 1959 (Activation: 0.5850)
  Words that maximally activate this feature:
  [' tweet', ' tweets', ' Tweet', 'tweet', ' tweeting']
  ↳ Low-Level Feature: 24 (Activation: 0.00011509)
     Words that maximally activate this low-level feature:
     [' tweeting', ' twitter', ' Twitter', ' TWITTER', ' twitter']

◆ Top-Level Feature 1939 (Activation: 0.3562)
  Words that maximally activate this feature:
  [' YouTube', ' Youtube', ' youtube', ' Google', ' YouTube']
  ↳ Low-Level Feature: 7 (Activation: -0.00020170)
     Words that maximally activate this low-level feature:
     [' goog', ' Flickr', 'yahoo', 'Wikipedia', 'Wiki']

◆ Top-Level Feature 506 (Activation: 0.1397)
  Words that maximally activate this feature:
  [' social', ' Social', 'Social', 'social', ' SOCIAL']
  ↳ Low-Level Feature: 3 (Activation: -0.00001523)
     Words that maximally activate this low-level feature:
     [' socialists', ' socialist', ' Socialist', ' Socialists', ' socialism']

◆ Top-Level Feature 1136 (Activation: 0.0638)
  Words that maximally activate this feature:
  [' blockchain', ' Blockchain', ' NFTs', ' TikTok', 'Blockchain']
  ↳ Low-Level Feature: 27 (Activation: 0.00007173)
     Words that maximally activate this low-level feature:
     ['新冠', ' Trump', 'Trump', ' Biden', ' https']

◆ Top-Level Feature 2046 (Activation: 0.0511)
  Words that maximally activate this feature:
  [' Tuesday', ' Wednesday', ' Thursday', ' Monday', ' Friday']
  ↳ Low-Level Feature: 22 (Activation: 0.00005836)
     Words that maximally activate this low-level feature:
     [' Sunday', 'Sunday', ' sunday', ' SUNDAY', 'SUNDAY']
```

(a) Baseline

```
Explanations for word: 'Twitter'
=================================================

◆ Top-Level Feature 1902 (Activation: 0.2157)
  Words that maximally activate this feature:
  [' gennaio', ' giugno', ' dicembre', ' settembre', ' />\r']
  ↳ Low-Level Feature: 11 (Activation: 0.25783429)
     Words that maximally activate this low-level feature:
     [' gennaio', ' giugno', ' dicembre', ' settembre', ' />\r']

◆ Top-Level Feature 799 (Activation: 0.1427)
  Words that maximally activate this feature:
  ['|,', '}`,', '>",', 'coders', 'dtype']
  ↳ Low-Level Feature: 24 (Activation: 0.29527846)
     Words that maximally activate this low-level feature:
     ['}`,', 'coders', '|,', ' legger', 'mec']

◆ Top-Level Feature 1473 (Activation: 0.1222)
  Words that maximally activate this feature:
  [' [\r', '2', '9', ' ={', '3']
  ↳ Low-Level Feature: 14 (Activation: 0.14846489)
     Words that maximally activate this low-level feature:
     [' [\r', ' ={', '4', ' [\r', '3']

◆ Top-Level Feature 1356 (Activation: 0.1189)
  Words that maximally activate this feature:
  ['ꝺu', 'zǒ', ' czego', 'futures', 'ㄱ']
  ↳ Low-Level Feature: 25 (Activation: 0.05206553)
     Words that maximally activate this low-level feature:
     [' Baseball', 'zǒ', ' बदल', ' UC', ' sacrament']

◆ Top-Level Feature 907 (Activation: 0.1167)
  Words that maximally activate this feature:
  ['ullah', ' threatening']
  ↳ Low-Level Feature: 24 (Activation: 0.12262511)
     Words that maximally activate this low-level feature:
     [' threatening', 'ullah']
```

(b) No Whitening

Figure 18: The causal inner product is necessary for the model to learn meaningful features.

```
Explanations for word: 'python'
=================================================

◆ Top-Level Feature 547 (Activation: 0.0955)
  Words that maximally activate this feature:
  [' pit', ' Pit', 'Pit', 'pit', ' pits']
  ↳ Low-Level Feature: 9 (Activation: 0.07605679)
     Words that maximally activate this low-level feature:
     [' py', 'Python', ' Python', ' python', 'python']

◆ Top-Level Feature 734 (Activation: 0.0839)
  Words that maximally activate this feature:
  [' script', ' Script', ' scripts', 'Script', 'script']
  ↳ Low-Level Feature: 4 (Activation: 0.00004199)
     Words that maximally activate this low-level feature:
     ['Script']

◆ Top-Level Feature 51 (Activation: 0.0826)
  Words that maximally activate this feature:
  [' snap', ' snaps', ' Sna', 'sna', 'snap']
  ↳ Low-Level Feature: 16 (Activation: -0.00001486)
     Words that maximally activate this low-level feature:
     ['Snap', ' snapping', ' SNA', ' Sni', ' sna']

◆ Top-Level Feature 1867 (Activation: 0.0624)
  Words that maximally activate this feature:
  [' Checkbox', ' DataBase', ' TextBox', ' mongodb', ' ToDo']
  ↳ Low-Level Feature: 19 (Activation: 0.06658750)
     Words that maximally activate this low-level feature:
     [' Javascript', ' javascript', ' JAVA', ' Kotlin', ' TypeScript']

◆ Top-Level Feature 1316 (Activation: 0.0506)
  Words that maximally activate this feature:
  [' ph', ' Ph', 'ph', 'Ph', ' PH']
  ↳ Low-Level Feature: 17 (Activation: 0.00003440)
     Words that maximally activate this low-level feature:
     [' Philip', ' Phillip', 'Philip', ' Philips', 'Phillip']
```

(a) Baseline

```
Explanations for word: 'python'
=================================================

◆ Top-Level Feature 1902 (Activation: 0.2456)
  Words that maximally activate this feature:
  [' gennaio', ' giugno', ' dicembre', ' settembre', ' />\r']
  ↳ Low-Level Feature: 24 (Activation: 0.29844666)
     Words that maximally activate this low-level feature:
     ['みました', 'ške', ' ^{', 'iology', '="/']

◆ Top-Level Feature 996 (Activation: 0.1957)
  Words that maximally activate this feature:
  ['ауı', 'ᴔ', ' ᴍ', 'んだろう', ' ᴄ']
  ↳ Low-Level Feature: 22 (Activation: 0.05954496)
     Words that maximally activate this low-level feature:
     ['cida', ' raps', ' dig', 'kenn', 'AEC']

◆ Top-Level Feature 696 (Activation: 0.1775)
  Words that maximally activate this feature:
  ['的不同', '립', 'ריח', '▷', 'firefox']
  ↳ Low-Level Feature: 20 (Activation: 0.17208365)
     Words that maximally activate this low-level feature:
     ['립', '립', ' vuelto', '▷', '的不同']

◆ Top-Level Feature 330 (Activation: 0.1550)
  Words that maximally activate this feature:
  ['ə', 'ują', ' strlen', 'utm', 'vať']
  ↳ Low-Level Feature: 15 (Activation: 0.25827643)
     Words that maximally activate this low-level feature:
     ['сów', 'categorical', 'чики', 'ул', 'vať']

◆ Top-Level Feature 145 (Activation: 0.1477)
  Words that maximally activate this feature:
  [', 《', 'дный', '場に', '体が', '嶋']
  ↳ Low-Level Feature: 12 (Activation: 0.10202558)
     Words that maximally activate this low-level feature:
     [', 《', ' K', 'ffed', ' gm', ' regresa']
```

(b) No Whitening

Figure 19: The causal inner product is necessary for the model to learn meaningful features.

```
Explanations for word: 'Bayesian'
==================================================
◆ Top-Level Feature 901 (Activation: 0.1695)
  Words that maximally activate this feature:
  [' Jacobian', ' Dirichlet', ' bilinear', ' piecewise', ' variational']
  ↳ Low-Level Feature: 31 (Activation: 0.00010277)
    Words that maximally activate this low-level feature:
    [' singularities', ' oscillatory', ' moduli', ' sinusoidal', ' trigonometric']

◆ Top-Level Feature 2010 (Activation: 0.1214)
  Words that maximally activate this feature:
  [' probability', ' probabilities', ' Probability', 'probability', ' Probab']
  ↳ Low-Level Feature: 15 (Activation: 0.00001307)
    Words that maximally activate this low-level feature:
    [' prospects', 'Probable', ' probable', ' Probable', 'probable']

◆ Top-Level Feature 870 (Activation: 0.0941)
  Words that maximally activate this feature:
  [' Catholic', ' Hindu', 'Catholic', ' Muslim', ' Christian']
  ↳ Low-Level Feature: 31 (Activation: -0.0001035)
    Words that maximally activate this low-level feature:
    [' Aryan', ' Huguen', ' Aryan', 'Gothic', ' Gregorian']

◆ Top-Level Feature 1837 (Activation: 0.0801)
  Words that maximally activate this feature:
  [' ba', ' Ba', 'Ba', 'ba', ' BA']
  ↳ Low-Level Feature: 30 (Activation: 0.07795604)
    Words that maximally activate this low-level feature:
    [' Bá', ' bay', ' Bay', 'Bay', ' BAY']

◆ Top-Level Feature 404 (Activation: 0.0781)
  Words that maximally activate this feature:
  [' statistics', ' Statistics', ' statistical', ' statistic', ' stats']
  ↳ Low-Level Feature: 12 (Activation: 0.00004004)
    Words that maximally activate this low-level feature:
    [' thống', ' stat', 'Estad', ' statistics', ' Statistical']
```

```
Explanations for word: 'Bayesian'
==================================================
◆ Top-Level Feature 1404 (Activation: 0.1600)
  Words that maximally activate this feature:
  ['hli', 'гово', 'Ontology', 'ufen', 'resolve']
  ↳ Low-Level Feature: 16 (Activation: 0.11451641)
    Words that maximally activate this low-level feature:
    ['Numbers', ' 官网', ' Verg', '这一刻', ' Fass']

◆ Top-Level Feature 1419 (Activation: 0.1000)
  Words that maximally activate this feature:
  ['くらい', '….', 'jte', 'welling', 'Très']
  ↳ Low-Level Feature: 5 (Activation: 0.03855699)
    Words that maximally activate this low-level feature:
    [' Onion', 'orang', ' ПО', ' Also', ' liaison']

◆ Top-Level Feature 652 (Activation: 0.0995)
  Words that maximally activate this feature:
  ['\\\\%', 'namespace', '\t\t\t', 'Banyak', ');\r']
  ↳ Low-Level Feature: 18 (Activation: 0.08977638)
    Words that maximally activate this low-level feature:
    ['));', 'вета', '))', 'apartment', ' );']

◆ Top-Level Feature 431 (Activation: 0.0880)
  Words that maximally activate this feature:
  ['ithi', 'ì', 'Root', 'fti', ' 재']
  ↳ Low-Level Feature: 31 (Activation: 0.00011427)
    Words that maximally activate this low-level feature:
    ['HET', ' Vintage', ' mẹ', ' correctional', ' home']

◆ Top-Level Feature 223 (Activation: 0.0843)
  Words that maximally activate this feature:
  ['public', 'Tính', '>', 'やき', 'Abstra']
  ↳ Low-Level Feature: 12 (Activation: 0.05679602)
    Words that maximally activate this low-level feature:
    [' Peoria', 'verk', 'public', ' ฐ', 'EF']
```

(a) Baseline

(b) No Whitening

Figure 20: The causal inner product is necessary for the model to learn meaningful features.

# C    MORE EXAMPLE FEATURES

**Expert #4073**

997)_**What**_would_become_Fuel_was_formed

_one_of_the_hosts_for_**What**_Now._Taylor_began_her

_Angelito",_"Lo_**Que**_No_Sabés_Tú",

-"Pie_Pod"_from_**What**_Would_You_Do?

_her_Sweet_Affliction_and_**What**_Remains_Secret_series._Born

**Sublatent 1**

52)_"**What**'s_on_the_Box

4:02_**What**_Is_Love?_3

_The_B-side_"**What**'s_More_(I

_"Pie_Pod"_from_**What**_Would_You_Do?

_Jay._The_line_"**What**_a_show,_there_they

**Sublatent 4**

It_includes_the_track_"**Where**_I_Find_My_Heaven",

_reworked_during_the_sessions_for_**Where**_Did_the_Night_

_In_his_book,_**Where**_the_Suckers_Moon,

:48_"**Where**_the_Sun_Don't

_Records_and_debut_**Where**'s_The_Beef?

**Sublatent 11**

_an_IT_consultant._In_**Who**_is_Review_Raja?_he

2018),_**Who**'s_Jenna_(2

_Cale;_they_include_"_**Who**_Knew",_"_Former_Me

_the_Jobs_of_Tomorrow,_**Who**_Gets_In_and_Why:

_inspiring_lines_such_as_"_**Who**_is_gonna_make_it/

Figure 21: An example of a hierarchical feature from the 8K highlevel x 16 sublatent H-SAE model. A 'question-word' high level feature with "What", "Where" and "Who" sublatents.

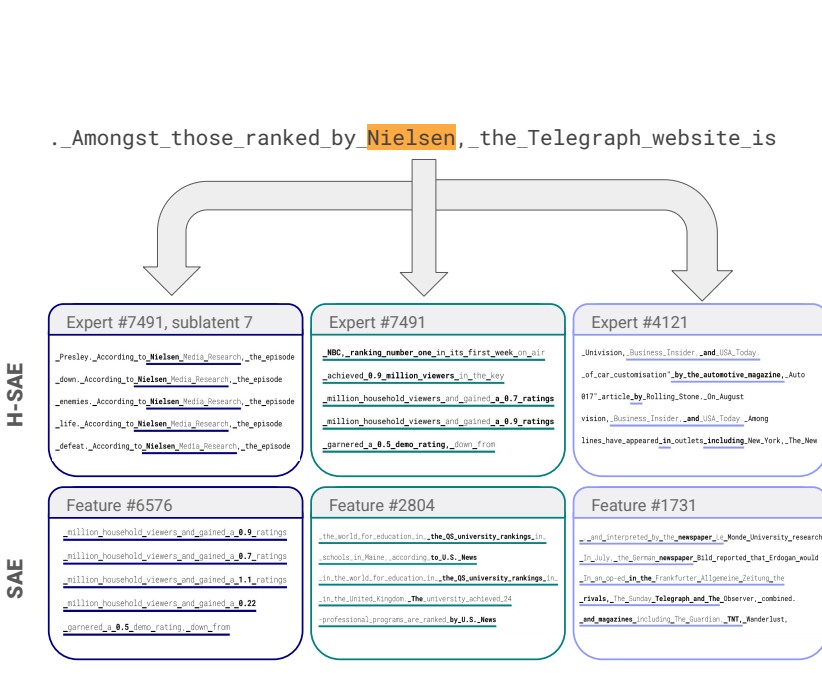

Figure 22: A comparison decomposition of the same context/embedding in the 8k x 16 H-SAE architecture and 8k standard SAE. Of note in the H-SAE is Expert #7491 which not only has the meaningful sublatent visualized here but also additional sublatents for various parts of writing about rating like demographics, specific ratings agencies like Nielsen, and mediums like TV episodes. Expert #4121 is a "magazines"/"news website" feature, promoting the following tokens heavily if added to the residual stream: _Forbes, _Bloomberg, _CNN, _BBC, _Daily, _Newsweek, _Reuters, _Huffington, _magazine, _The. We see a similar decomposition on the SAE, with a newspaper feature and "media ratings" feature. However, due to the lack of granular features, the specific ratings agency 'Nielsen' is not represented (rather the top-level ratings feature promotes the token 'Nielsen' more heavily in the SAE). The SAE uses a more generic "ratings"/"rankings" feature that shows signs of absorption with a "University Rankings" feature.

# D    COMPARISONS WITH MATROYSHKA-SAE

## D.1    SYNTHETIC BENCHMARK

The M-SAE paper introduces a synthetic benchmark that consists of 20 features organized hierarchically into 11 "parent features" (which can co-activate) and 9 child features split evenly among 3 of the parents (so 8 "parent" features have no children). We adapt this to 28 overall, with 12 parents, and 16 children split among 4 parents. Following the Matrysoka paper, we train a M-SAE with 28 features. We also train a 12x4 H-SAE (notice that this model is misspecified for the data because we allow all parents to have children). The M-SAE paper evaluates the reconstruction by cosine similarity between the optimal alignment of the learned and ground truth features. We find the M-SAE achieves a mean cosine similarity of 0.502 and the H-SAE 0.526 (despite the H-SAE being misspecified)

## D.2    UNEMBEDDING DECOMPOSITION

We train an Hierarchical-SAE and Matroyshka-SAE for 10,000 steps on the unembeddings transformed by the causal inner product. The H-SAE is a top-k=5, 2k x 32 model and the M-SAE is matched with top-k=10 and 5 groups out of 65k total features (i.e. both models have the same number of overall latents and can select the same number of latents per reconstruction). The features are fairly similar, with the H-SAE often finding sparser representations even within its top-k (by having near-zero activations on irrelevant features), finding high-level features the M-SAE does not, and fewer uninterpretable features. The lower maximum activation may also be an advantage of the H-SAE, as it spreads its activation weight out over more composable features. These factors also combine with the reconstruction loss and computational advantage the H-SAE has over both an M-SAE and standard SAE (e.g. the H-SAE computes 3% as many feature activations during the encoding stage as compared to the M-SAE). Comparison decompositions follow:

## Decomposition of ' Chicago'

| Feature Num | | Activation | | Top Tokens | |
| --- | --- | --- | --- | --- | --- |
| H-SAE | M-SAE | H-SAE | M-SAE | H-SAE | M-SAE |
| 1927 | 1368 | 0.4056 | 0.5899 | [' Toronto', ' Chicago', ' Atlanta', ' Mumbai', ' Denver'] | [' Chicago', 'Chicago', ' CHICAGO', ' chicago', 'chicago'] |
| 1927.5 | 737 | 1E-3 | 0.4034 | [' Madrid', 'Madrid', ' Barcelona', 'Barcelona', ' Tucson'] | [' Seattle', ' Atlanta', ' Detroit', ' Denver', ' Chicago'] |
| 859 | 44230 | 0.1282 | 0.0462 | [' American', ' America', ' american', 'American', ' Americans'] | ['chicago', 'ந்து', ' chicago', 'ந்த', ' Chicago'] |
| 859.5 | 37392 | 3E-4 | 0.0164 | [' USA', 'USA', ' الايا', ' EUA', ' США'] | [' Missouri', ' Wisconsin', ' Indianapolis', ' Kansas', ' Tennessee'] |
| 175 | 15282 | 0.1170 | 0.0403 | [' Texas', ' Alabama', ' Florida', ' Louisiana', 'Texas'] | [' Earth', ' West', ' South', ' King', ' Bay'] |
| 175.19 | 3961 | 0.0522 | 0.0525 | ['iowa', ' Iowa', ' Kansas', 'Iowa', 'Kansas'] | [' Pennsylvania', ' Michigan', ' Illinois', ' Wisconsin', ' Ohio'] |
| 98 | 4981 | 0.1094 | 0.0329 | [' Chrom', ' chrom', 'chrom', ' Chrome', 'Chrom'] | ['NYC', ' NY', ' NYC', ' NY', ' nyc'] |
| 98.0 | 34993 | -2E-4 | 0.0069 | [' Chromosome', 'Chromosome', ' browser', ' thermo', ' nhiễm'] | [' MEXICO', 'mexico', ' mexico', 'Mexico', ' Mexico'] |
| 1512 | 64996 | 0.0887 | 0.0067 | [' il', 'il', ' Il', ' IL', 'Il'] | ['ới', ' giới', 'ời', 'Amore', ' }}{\\'] |
| 1512.8 | 34031 | 1.6E-3 | 0.0066 | [' ila', ' Illusion', ' ilo', 'ilation', 'ыл'] | [' rör', ' haberse', ' haber', ' skydd', ' cc'] |

## Decomposition of ' puppy'

| Feature Num | | Activation | | Top Tokens | |
| --- | --- | --- | --- | --- | --- |
| H-SAE | M-SAE | H-SAE | M-SAE | H-SAE | M-SAE |
| 732 | 282 | 0.3964 | 0.6628 | [' dog', ' Dog', 'Dog', 'dog', ' dogs'] | [' puppy', ' Puppy', 'Puppy', 'puppy', ' puppies'] |
| 732.31 | 177 | -0.0006 | 0.3691 | [' barks', ' собаки', ' woof', ' собака', ' Labrador'] | [' dog', ' dogs', ' Dog', 'dog', 'Dog'] |
| 956 | 27992 | 0.0883 | 0.0356 | [' pig', ' goat', ' pigs', ' monkey', 'Goat'] | [' brother', ' mother', ' father', ' sister', ' son'] |
| 956.12 | 3008 | 9E-5 | 0.0633 | [' horse', 'Horse', ' Horse', ' pony', 'Pony'] | [' infant', ' Infant', ' infants', ' baby', ' babies'] |
| 117 | 13729 | 0.0525 | 0.0361 | [' boy', ' Boy', 'boy', 'Boy', ' BOY'] | [' wood', ' fish', ' glass', ' milk', ' snow'] |
| 117.0 | 59925 | 0.2141 | 0.0068 | [' menina', ' boya', 'ガール', 'BOYS', 'GIRLS'] | ['\uf075', ' \uf075', ' zamanda', '玩笑', '\uf06e'] |
| 487 | 46772 | 0.0457 | 0.0067 | [' pet', ' Pet', 'pet', 'Pet', ' PET'] | ['เยอะ', 'มากๆ', ' น', ' oxpa', 'เขียน'] |
| 487.4 | 6188 | 1E-4 | 0.0264 | ['PETER', ' Pedro', 'Pedro', 'petrol', 'Petrol'] | [' Teddy', 'Teddy', 'teddy', ' teddy', ' Ted'] |
| 1809 | 36597 | 0.0437 | 0.0065 | [' patient', ' shopper', ' attendee', ' subscriber', ' sufferer'] | ['ด', ' ふる', 'จด', ' inau', '獣'] |
| 1809.8 | 63546 | 9E-4 | 0.0065 | [' Passenger', ' passenger', 'passenger', 'Passenger', ' guest'] | ['MDL', ' MDL', 'mdl', 'NOV', 'MRP'] |

**Decomposition of 'Queen'**

| Feature Num | | Activation | | Top Tokens | |
|---|---|---|---|---|---|
| H-SAE | M-SAE | H-SAE | M-SAE | H-SAE | M-SAE |
| 951 | 99 | 0.5322 | 0.8691 | [' King', ' king', 'King', ' kings', ' KING'] | [' Queen', 'Queen', ' queen', ' QUEEN', ' queens'] |
| 951.24 | 2304 | -9E-3 | 0.1364 | [' crowns', 'könig', ' crowns', ' crown', ' roy'] | [' Coach', ' Owner', ' Officer', ' Manager', ' Administrator'] |
| 1724 | 15282 | 0.2876 | 0.0816 | [' lady', ' woman', ' Lady', 'lady', 'Lady'] | [' Earth', ' West', ' South', ' King', ' Bay'] |
| 1724.15 | 2304 | 7E-4 | 0.0109 | ['girlfriend', ' girlfriend', 'Girlfriend', ' girlfriends', ' madam'] | [' Staten', '諫', ' Axes', ' Rhode', 'Rhode'] |
| 1247 | 43216 | 0.2281 | 0.0076 | [' Qu', ' qu', 'Qu', ' QU', 'qu'] | ['goers', ' Watcher', 'qc', 'QC', ' VIEWS'] |
| 1247.13 | 30719 | 1.3E-3 | 0.0147 | ['QUEUE', ' Que', 'Que', ' QUE', 'QUE'] | ['Jerusalem', ' Jerusalem', ' القدس', 'lande', ' Haifa'] |
| 1914 | 53291 | 0.1337 | 0.0073 | [' Officer', ' Engineer', ' Professor', ' Trainer', ' Surgeon'] | ['믜', ' 믜', ' ϻ', ' ϻ', '�灾'] |
| 1914.29 | 34322 | 4E-4 | 0.0073 | [' Presidents', ' Blogger', ' 威', ' Maestro', ' Appellant'] | ['寻求', ' recibido', ' kuitenkin', 'medal', '洵'] |
| 939 | 45011 | 0.0612 | 0.0073 | [' chairman', ' Chairman', 'chairman', 'Chairman', ' CEO'] | [' 伸', '伸', ' Bronson', '肘', 'レッド'] |
| 939.8 | 59557 | 5E-3 | 0.0071 | [' President', ' president', 'President', 'president', ' PRESIDENT'] | [' VTT', ' Norsk', ' Matti', ' formant', ' Marat'] |

**Decomposition of 'Bayesian'**

| Feature Num | | Activation | | Top Tokens | |
|---|---|---|---|---|---|
| H-SAE | M-SAE | H-SAE | M-SAE | H-SAE | M-SAE |
| 901 | 8866 | 0.1695 | 0.4217 | [' Jacobian', ' Dirichlet', ' bilinear', ' piecewise', ' variational'] | ['Bayesian', ' Bayesian', ' Bayes', 'Bayes', ' bayonet'] |
| 901.32 | 9192 | 1E-3 | 0.1400 | [' singularities', ' oscillatory', ' moduli', ' sinusoidal', ' trigonometric'] | ['Probability', ' Probability', 'probability', ' probability', '概率'] |
| 2010 | 2552 | 0.1214 | 0.2152 | [' probability', ' probabilities', 'Probability', 'probability', ' Probab'] | [' Dirichlet', ' Jacobian', ' Cauchy', ' Laplace', ' Poincaré'] |
| 2010.15 | 3286 | 1E-4 | 0.1930 | [' prospects', 'Probable', ' probable', 'Probable', 'probable'] | ['ay', 'AY', ' Ay', ' ay', 'Ay'] |
| 870 | 858 | 0.0941 | 0.1309 | [' Catholic', ' Hindu', 'Catholic', 'Muslim', ' Christian'] | [' ba', ' Ba', 'Ba', 'ba', ' BA'] |
| 870.31 | 1395 | -1E-4 | 0.686 | [' Aryan', ' Huguen', ' Aryan', 'Gothic', ' Gregorian'] | [' probability', ' likelihood', ' chances', ' probabilities', ' Probability'] |
| 1837 | 1367 | 0.0801 | 0.0593 | [' ba', ' Ba', 'Ba', 'ba', ' BA'] | [' Catholic', 'Catholic', ' Catholics', ' catholic', ' Muslim'] |
| 1837.30 | 5299 | 0.0780 | 0.0459 | [' Bá', ' bay', ' Bay', 'Bay', ' BAY'] | [' statistics', ' statistical', ' Statistics', 'Statistics', 'statistics'] |
| 404 | 57380 | 0.0781 | 0.0099 | [' statistics', ' Statistics', 'statistical', ' statistic', ' stats'] | ['広場', ' priors', '样本', 'Bayesian', ' YC'] |
| 404.12 | 39412 | 4E-4 | 0.0085 | [' thống', ' stat', 'Estad', ' statistics', ' Statistical'] | [' epistem', ' methodological', ' Epis', 'EPISODE', ' evangel'] |