# OpenReview forum: "Incorporating Hierarchical Semantics in Sparse Autoencoder Architectures"
_ICLR.cc/2026/Conference — Submitted to ICLR 2026_

### Official Review · Reviewer_iFYM · 2025-10-22

**Soundness:** 1
**Presentation:** 2
**Contribution:** 2
**Rating:** 2
**Confidence:** 4

**Summary:**

This work introduces and evaluates a new Sparse Autoencoder (SAE) architecture designed to capture Hierarchical relations in the data. This method brings a mixture of experts approach to an architecture similar to Matryoshka-SAEs. The authors explore the interpretability of the features recovered by this new method and compare it to standard SAEs along commonly used metrics.

**Strengths:**

- Adressing the limitations of SAES is an important line of research
- Combining ideas from mixture of experts with an approach like Matryoshka-SAE seems like a promissing direction.

**Weaknesses:**

Unfortunately, the paper fails to make a convincing case for the advantages of H-SAE over existing methods due to the following limitations of the evaluation methodology.

**Sparsity as a confounder:** The authors use metrics from SAEBench but do not incorporate the following evaluation practice prescribed in the SAEBench github page. This makes it unclear whether H-SAE improves performance at the same sparsity level, plots for the key metrics over sparsity levels at the same dictionary size seems crucial to establish the advantages of H-SAE.

>'Recommended Evaluation Practice
When evaluating new SAE methods, we strongly recommend training multiple SAEs across a range of sparsities (e.g. L0 ∈ [20, 200]) alongside directly comparable baselines. Many evaluation metrics correlate strongly with sparsity, so assessing performance across multiple sparsity levels is essential to avoid misleading conclusions.

>This practice helps ensure that observed improvements are real and not just artifacts of sparsity or statistical noise. It also makes it easier to determine whether a method actually improves the Pareto frontier on target metrics.'

Very limited comparison to the most similar existing method (Matryoshka-SAE). Figure 4 would be more convincing if it were a comparison with Matryoshka-SAE or at least BatchTopK-SAE at the same sparsity level with other examples in the Appendix to show this one isn't cherry picked.

**Questions:**

I think the H-SAE approach is reasonable, but the evaluation of the method has severe limitations. To increase my score I would need to see:

1) Plots comparing existing SAEs (ideally including Matryoshka and BatchTopK) to H-SAE along different sparsity levels for the same dictionary size.
2) Reporting of sparsity levels for the different SAEs in the current results/figures and making sure they are similar for a fair comparison. If the sparsity levels are different, the l1 coefficient or topk values should be adjusted to line them up.

Beyond this I would also like to see a more detailed evaluation and discussion of the pros and cons of this method when compared to existing SAEs. While I think the approach is interesting and worth pursuing, given the magnitude of the changes needed in the evaluation I would recommend taking some time to improve the paper and resubmitting rather than trying to make all these changes in the rebuttal, but I could change my mind if new convincing evaluations are provided.

---

> ### Author Response · Authors · 2025-12-04
>
> Thank you for your review\! We agree that it is important to address the drawbacks of SAEs.
>
> With respect to your concerns about comparison and interpretability:
>
> 1. Our response to reviewer H1X7 has more details on what we think about our comparisons to the M-SAE. But, at a high level: The M-SAE changes are orthogonal to ours and both techniques could be combined\! The improvement of the H-SAE relative to baselines is (1) Incorporating hierarchical structure drastically improves computational efficacy (2) This structure also makes concepts easier to parse (e.g. ‘dog’ and ‘corgi’ features are explicitly connected). These advantages are by definition, so we’re testing mostly if the method actually works, and our baselines and evaluations are chosen to answer that question
> 2. We then see the key questions (that are also easier to answer empirically) as: (1) are the hierarchical arrangements discovered basically coherent (2) do we get any backsliding on non-hierarchical interpretability?
> 3. M-SAEs are more expensive to train than Top-K, so we only do comparisons on the word level to make sure we don’t have any backsliding as compared to an existing way of increasing interpretability, and find this to be true. In fact, the M-SAE changes are orthogonal to ours and both techniques could be combined. Similarly, a BatchTopK activation function can be combined with either the H-SAE or M-SAE
> 4. To help us answer these questions, we also work hard to enable qualitative evaluation without the typical cherrypicking concerns. We included notebooks where you can do your own visualization and sets of randomly selected feature activations to explore
>
> With respect to concerns about sparsity:
>
> 1. Because of our hierarchical structure, sparsity becomes harder to conceptualize. Our features are forced to co-activate (delivering inherent interpretability gains\!), making it hard to compare to flat sparsity and define it fairly. So we think treating the married pairs of features together is best. Alternatively, if we decouple our features and consider the "proportion of latents active,” the H-SAE delivers clear sparsity gains. Empirically, the low level features often do not activate meaningfully (they only activate when the more granular concept is relevant\!), so the sparsity advantage is even larger.
> 2. More broadly, in terms of sparsity scaling, we’ve run 5 new H-SAE training runs with 200 million tokens to observe sparsity scaling patterns, and they’re as expected (see plots here: [https://imgur.com/a/FaMj4ud](https://imgur.com/a/FaMj4ud)). As sparsity is decreased, reconstruction loss decreases as does the absorption fraction score, which is the observed SAE scaling behavior on SAEBench ([https://www.neuronpedia.org/sae-bench?modelId=gemma-2-2b\&layer=12\&dSae=65536\&release=gemmascope%2Csae\_bench\&metricX=core%7C%7Csparsity%7C%7Cl0\&metricY=tpp%7C%7Ctpp\_metrics%7C%7Ctpp\_threshold\_20\_total\_metric\&groupBy=saeClass](https://www.neuronpedia.org/sae-bench?modelId=gemma-2-2b&layer=12&dSae=65536&release=gemmascope%2Csae_bench&metricX=core%7C%7Csparsity%7C%7Cl0&metricY=tpp%7C%7Ctpp_metrics%7C%7Ctpp_threshold_20_total_metric&groupBy=saeClass)). Notably, this is inverse to the M-SAE scaling dynamic ([https://www.neuronpedia.org/sae-bench?modelId=gemma-2-2b\&layer=12\&dSae=65536\&release=gemmascope%2Csae\_bench\&metricX=core%7C%7Csparsity%7C%7Cl0\&metricY=absorption\_first\_letter%7C%7Cmean%7C%7Cmean\_full\_absorption\_score\&groupBy=saeClass](https://www.neuronpedia.org/sae-bench?modelId=gemma-2-2b&layer=12&dSae=65536&release=gemmascope%2Csae_bench&metricX=core%7C%7Csparsity%7C%7Cl0&metricY=absorption_first_letter%7C%7Cmean%7C%7Cmean_full_absorption_score&groupBy=saeClass)), but it is unclear a priori which should be preferred.

---

### Official Review · Reviewer_Y3qo · 2025-10-26

**Soundness:** 3
**Presentation:** 3
**Contribution:** 3
**Rating:** 4
**Confidence:** 3

**Summary:**

This paper introduces a hierarchical sparse autoencoder approach.  This includes a novel architecture and 3 tweaks to the training process (which are tested in ablations in the appendix), and is inspired by "The Geometry of Categorical and Hierarchical Concepts in Large Language Models" - Park et al., ICLR 2025.

The basic idea is this:
- There is one top-level SAE, and many low-level SAEs, one for each feature in the top-level SAE.
- You first run the top-level SAE, and see which features are active.
- Then for each active feature, you run the corresponding low-level SAEs.
- The reconstruction of an input is given by the sum of the outputs of all of the (top-level and low-level) SAEs that were run on that input.

The low-level SAEs also include projection matrices for dimensionality reduction(/expansion).

The authors highlight:
- The conceptual motivation, novelty, and significance.
- Improvements in reconstruction and interpretability.
- Computational savings.

**Strengths:**

I find the contribution of the paper compelling.  It seems well-motivated, significant, and effective.
The lack of feature hierarchy in SAEs does seem like an important issue, especially in light of Park et al. (2025).

The paper is quite clear and well-executed overall.  The main exception to this is that Matryoshka SAEs are not given enough attention; this is a critical weakness to me at the moment, since I think it's essential to clearly demonstrate advantages and novelty over the most closely related work.

**Weaknesses:**

- (major) I’d like to see the Matryoshka SAE featured more prominently.  It's plausibly a more important point of comparison than a standard SAE.  So I'm surprised to see the experiments being kept to the appendix and being more limited (e.g. no comparison in terms of explained variance).  I would also usually expect it to be introduced in sufficient detail to understand how it works without reading their paper (e.g. including key equations).  And I think the claim that H-SAEs have “fewer uninterpretable features” needs to be explicitly supported when it is made.  I’m also not sure if the comparison between the two methods is fair in terms of parameters and compute; this should be clarified.
- (minor): I'd like to understand the connection with Park et al. more thoroughly,
- (minor): Incorrect characterization of background / related work in the first sentence of the abstract: “Sparse dictionary learning (and, in particular, sparse autoencoders) attempts to learn a set of human-understandable concepts that can explain variation on an abstract
space.”  But this is not necessarily the goal of sparse dictionary learning.
- (minor): I'm unconvinced by the way this work is framed as related to causal representation learning.  I’m not aware of any reason to expect the features learned by a sparse autoencoder to be causal.
- (minor): “In particular, they show that every categorical concept” I think “show” is a bit strong here; I believe this is in the context of a particular formalism, and it’s unclear the extent to which LLMs universally mirror it empirically.
- (nit): “feature absorption” is not defined

(just FYI) Some typos:
- “We also highlight Switch SAEs (Mudide et al., 2024), which introduce a mixture-of-experts type routing mechanism shares some structural similarities to our approach.”
- citeps in first paragraph of related work
- “The H-SAE with 64 sublatents per expert is on par with the standard SAE with 32 top-level features” → “The H-SAE with 8k top-level features and 64 sublatents per expert is on par with the standard SAE with 32k top-level features”

**Questions:**

While the work stresses computational efficiency gain, it seems like it also potentially increases the number of parameters, which is worth noting.  Relatedly, I’m not quite convinced by the claim that: “because the memory cost of a batch gradient step scales with the number of activated parameters, not the total number of parameters, this efficiency also applies to (per-step) training.”  Indeed, within a (large) batch, I’d expect many different top-level atoms to be active, requiring the loading of many low-level SAEs.  Furthermore, I’m not sure if parameters can be moved on and off GPU quickly enough to make this practically useful (e.g. if memory is the bottleneck on your GPU, then I think you would need to reduce the number of top-level atoms or move low-level SAE’s parameters on and off the GPU with every batch). I’d like to see versions of Figure 3 with parameters and compute cost on the x-axis.

An important limitation which I don’t see mentioned is that the model is restricted to a 2-level hierarchy.  Would it be straightforward to extend to arbitrary depth?

Does eqn (1) really need a LeakyReLU in it?  Won’t the top-K activations typically be positive anyhow?

---

> ### Author Response · Authors · 2025-11-30
>
> Thank you for your review\! We’re glad you find our method significant and effective.
>
> 1. (Comparison with M-SAE). Our response to reviewer H1X7 has more details on what we think about our comparisons to the M-SAE. But, at a high level: The M-SAE changes are orthogonal to ours and both techniques could be combined\! The improvement of the H-SAE relative to baselines is (1) Incorporating hierarchical structure drastically improves computational efficacy (2) This structure also makes concepts easier to parse (e.g. ‘dog’ and ‘corgi’ features are explicitly connected). These advantages are by definition, so we’re testing mostly if the method actually works, and our baselines and evaluations are chosen to answer that question
>
> With respect to some other concerns:
>
> 1. (multiple levels of hierarchy) Our method would expand to more levels of hierarchy cleanly, see our response to reviewer gqjX on this concern\! The main question is how many levels to model explicitly.
> 2. (memory and compute cost) Our scaling with respect to memory footprint will be worse, but we think this trade-off is hard to articulate fully. In terms of the practical considerations of SAE training, they are extremely shallow models with few parameters–our largest model has only 1.5GB of weights (most of the engineering effort is in getting data to GPUs fast enough to not completely bottleneck training–OpenAI and Google both talk about this in their SAE papers\!). So, we’re far more constrained by (1) The flop intensity of each forward pass (2) The data bandwidth for each forward pass. The H-SAE significantly reduces this flop intensity, which we think is the relevant factor for scaling. In the end we train an SAE with millions of overall latents (32k x 64), the same scale OpenAI/Anthropic have, with a fraction of the compute\!

---

### Official Review · Reviewer_U78n · 2025-10-27

**Soundness:** 4
**Presentation:** 3
**Contribution:** 3
**Rating:** 4
**Confidence:** 5

**Summary:**

In this work, the authors propose a new architecture for Sparse Autoencoders to extract hierarchical concepts from data. They draw from previous theories on the hierarchical representation of concepts in LLMs to design an architecture with an inductive bias to find high-level parent features and low-level children features. Specifically, this is instantiated as a mixture-of-experts approach where a Top-K SAE finds the high level concepts present in a representation and then selective activates low-level Top-1 SAEs which determine the specific child feature pertains to the input.

**Strengths:**

* The authors provide a novel architecture and approach grounded in theories of hierarchical concept representations in LLMs
 * The method is simple and intuitive
 * Empirical results highlight the efficacy of the method in terms of reconstruction metrics as well as more nuanced benchmarks such as feature absorption and feature universality across language.
 * I think the absorption experiment is a great way to demonstrate the efficacy of this method, as it intuitively (and empirically) seems clear that hierarchy should help with the splitting problem.

**Weaknesses:**

(Apologies for any clarity issues, I bounce between saying high- and low-, top- and low-, etc. to describe your hierarchy of features).

The authors argue that this architecture is more useful for three primary reasons: 1) H-SAEs have better reconstruction, 2) H-SAEs are more interpretable, and 3) H-SAEs learn hierarchical semantics. However, I see some issues with these claims:
1. If I understand Figure 3 properly, you compare H-SAEs against SAEs with the same number of top- or total number of latents respectively. This means that, with the "expert" sub-SAEs, the H-SAEs have 32 or 64x as many parameters as compared to an SAE, which I believe is an unfair comparison. I think it would be more interesting if you can show that given the same computational or parameter budget, reorganizing your SAE into a hierarchical structure is better. This would convince me more strongly that this inductive bias is the right way to learn the representation structure as it would demonstrate it is more efficient at learning and better at reconstruction. Instead, I can't be sure that these gains in performance simply come from using a bigger model.
2. Its hard for me to interpret the quality of the numbers in Fig. 5b. If we are looking at set difference, it seems to me that the both SAEs differ on a significant proportion of features. If I were to frame it another way, out of the top-8 features for both SAEs on different translations of the same sentence, I can expect about 4 to differ, which is significant. It seems as though a difference of about 7 versus 9 is minor compared to the fact that both SAEs clearly exhibit disagreements across languages. I appreciate the introduction of a new evaluation for feature redundancy, and would love your comments on what you expect as a 'good result' for this evaluation.
3. I agree that by design, we should see 3) and it appears that this is happening qualitatively. However, the performance of this is briefly addressed in a small paragraph on line 459 and not much is discussed beyond qualitative examples. I understand that a common difficulty in interpretability research is providing quantitative or large-scale evaluations, but I think this is still quite brief given it is the entire goal of the paper. Most of the experiments are devoted to reconstruction and the 2 interpretability benchmarks, but I feel like this is the most important component of the paper. From a practitioner's perspective, I guess I would pick to train an H-SAE over a normal SAE because of the reconstruction results, but from a scientific perspective, a much more interesting question is using your method to evaluate the veracity of the hierarchical concept representation hypothesis. How hierarchical are LLM representations in practice? If this hierarchy is the right model of LLM representations, how will we know and what can we do with it? Instead, this paper's contribution seems to mainly be bumping existing metrics up with some qualitative examples of why it is better.

Other comments:
 * Additional loss terms: The discussion from lines 300-310 confused me a little bit. I read it as "we did X and Y because we think it is good, but its not clear if it actually is." Reading this part weakened the messaging of the paper, and it seems as though it was unnecessary. Maybe this just needs to be framed more confidently, but I think it might be better off removing these parts. When I read Algorithm 1, my first question was why you were including a sparsity loss given you were using top-k architectures, which don't use a sparsity penalty [1]. Later I understood that this is being used as an auxiliary loss term on the non-activating features, but it appears in Alg 1 to be applied to all features. Even so, why change from the usual auxiliary loss? It makes it harder to compare unless you can also run an ablation comparing these two design choices.
 * Efficiency claims: Similar to my point above on reconstruction metrics, how would the computational efficiency compare with an SAE trained with the same number of **total** parameters (so j + js features or just js features)?

Typos/Formatting:
* Line 196, it seems as though you added too many parentheses in z_j = (LeakyReLU(Ex))
* Figures 3 and 5 are formatted quite badly. You should remove the subcaptions and include them all in the overall caption as "Figure 5: **(a)**: The standard SAE... **(b)** The H-SAE...". Instead, you have 3 captions that are all aligned differently and run off the page. Additionally, you should try to align the left and right subfigures vertically, and avoid having them break the margins if possible.
 * On line 367, do you mean 32k and not 32 top-level features?
 * The citation for "Unlocking Hierarchical Concept Discovery in Language Models..." is in all caps.

Overall, I really like this idea and the proposed architecture, but the evaluations fall a little short and are closer to the minimum required to evaluate a novel SAE. I think this work has huge potential for interesting applications, evaluations of the method, or evaluations of the hierarchical concept representation hypothesis and I would be willing to improve my score if these were explored or addressed but that would require pretty large changes to the existing submission. I also apologize for the long-winded review, I am more than happy to clarify anything during the discussion period!

[1] Gao, L., la Tour, T. D., Tillman, H., Goh, G., Troll, R., Radford, A., ... & Wu, J. (2024). Scaling and evaluating sparse autoencoders.

**Questions:**

* Did you try having the low-level SAEs take in the residual only? I.e. instead of $SAE^j_1(\Pi x))$, doing  $SAE^j_1(\Pi (x-x_{high}))$.
 * I would be interested in seeing an ablation of Top-1 for the low-level SAEs. Why not try allowing for more low-level concepts? Maybe a representation contains information about a wedding AND a divorce. This relates to my point in the Weaknesses section -- this paper would be much more compelling with a thorough investigation of the hierarchical decompositions unlocked by your model and more evaluations or explorations of this component as opposed to the brief treatment they currently receive.
 * Additionally, I would be interested in understanding the hierarchical structure of the language features explored in the redundancy experiment. You implicitly hypothesize that the top-level feature would be a multilingual syntax feature, with the low-level features describing language difference, but what if it were the other way around? Specifically, a high-level language feature, each with its own low-level syntax feature. I find this discussion interesting and recent work has looked at the relationship between syntax and meaning in SAE features and notably specifically modeled syntax as a low-level feature [2]. I would appreciate a deeper dive into the structure that explains Figure 5b and how your hypothesized hierarchy informs your evaluation design. I would also appreciate an explicit description of your hypothesized hierarchy (although you do slightly touch on this in the last sentence on page 8).
 * On that note, one thing [2] does is they evaluate their high- and low-level features separately. I think this sort of investigation could also improve the work: Would the feature redundancy be much lower if just looking at top-level features? How does absorption change when looking at top features or sublatents or both? Are you doing all of the evaluations in this paper on top-level features only?
 * In footnote 1, when you say primal and dual space representations, what do you mean? Why is a dual being introduced?

[2]: Bhalla, U., Oesterling, A., Verdun, C. M., Calmon, F., & Lakkaraju, H. Leveraging the Sequential Nature of Language for Interpretability.

---

> ### Author Response · Authors · 2025-11-30
>
> Thank you for your review\! We’re glad you found the approach novel and intuitive and our absorption method a good way of measuring our results. Most broadly, we agree that the question is about hierarchical semantics. Given we know the basic structure from Park et al. we focus on explicitly modeling this hierarchy for inherent interpretability benefits, and careful qualitative evaluation to validate the method. With respect to some of your specific concerns:
>
> 1. (figure 3 comparisons) In traditional flat SAEs, total width, computational complexity, and the number of parameters all scale the same. A main advantage of the hierarchical structure is that these three things decouple. In fact, because of the MoE and projected subspace structure, in our actual experiments the additional computational cost and number of parameters added to account for the hierarchical structure is comparatively tiny. For example, in the experimental setting with k=32, proj\_dim=8, top\_level\_dim=2304, number\_of\_low\_level\_atoms\_per\_expert=64, and number of top level atoms 8k, the standard SAE has \~18.9 million active parameters, and the H-SAE adds on only 600,000. That is, moving from the standard SAE to the H-SAE increases parameter count by just 3%. In all other experimental settings, this increase is even smaller. (Changing the x-axis of the plots to be this value is virtually imperceptible). Despite this tiny increase, this model outperforms a 32k standard SAE with 113M more parameters on reconstruction: we think this decoupling is really powerful\!
>
> 2. (evaluations) One thing to emphasize here is that, relative to baseline methods, the main advantages of the H-SAE method are \\emph{by design}. Both the computational efficiency from the mixture structure and explicit grouping of related atoms just don’t exist in the comparison methods. Accordingly, what we’re actually trying to evaluate empirically here is mainly whether the H-SAE architecture does basically succeed at learning. Then, the metrics we’re measuring are mostly aimed at checking we aren’t suffering any performance degradation relative to baselines. In this view, the main results are the reconstruction performance and the qualitative examination showing that we do in fact learn hierarchical structure.
>
>    Of course, the major issue with qualitative evaluation is scale---it’s hard to know if examples are cherry picked. For this reason, the supplementary materials include visualization tools we developed to allow readers to check examples themselves (from among a very large randomly selected set). In this manner, we can get something like qualitative evaluation at scale.
>
>    The adsorption and cross-lingual feature sharing experiments are more in the flavor of scientific understanding of what effect the method has than directly in support of an engineering thesis. The takeaway is simply that there is an apparent quantitative improvement in compositionality. We do not have any claims about what a ‘good’ score would be in absolute terms. (Indeed, it’s not obvious a priori how much change we would \*ideally\* expect in representations when the language gets changed)
>
> 3. (loss terms) Our response to reviewer gqjX about the subspace dimension is informative here. In other words, we don’t find hyperparameter changes to have dramatic effects. We think the most important prerequisite research direction for maximal tuning  is finding the “right” geometry. We’re mostly reporting our choices for reproducibility, not as a main contribution\!

---

### Official Review · Reviewer_gqjX · 2025-10-30

**Soundness:** 4
**Presentation:** 4
**Contribution:** 3
**Rating:** 8
**Confidence:** 4

**Summary:**

This paper tackles the well-known reconstruction-interpretability trade-off in SAEs, which often suffer from feature splitting as they are scaled up. The authors propose a new architecture, the Hierarchical SAE, which explicitly models the hierarchical nature of semantic concepts. The architecture uses a MoE-like design: 1. A top-level SAE identifies general "parent" concepts. 2. The activation of these top-level features gates smaller, specialized "expert" SAEs. 3. These experts model "child" concepts by operating in a dedicated low-dimensional subspace. 4. The final reconstruction is a sum of the high- and low-level representations, directly implementing the "corgi" = "dog" + "corgi-in-dog-space" structure.

When applied to LLM internal activations, this H-SAE architecture improves reconstruction performance. It matches a SAE 4x its size while being more computationally efficient. It also learns more interpretable and less redundant features.

**Strengths:**

The core idea of using a hierarchical, MoE-style architecture for SAEs is simple but practical. The gain in computational efficiency is a significant result on its own, potentially unblocking efforts to scale interpretability tools to frontier models.

The experimental validation is quite thorough. The authors combined standard reconstruction loss with metrics for downstream task performance like CE loss, as well as feature absorption and cross-lingual feature redundancy. The direct comparison to Matryoshka SAEs on multiple tasks including their own synthetic benchmark is convincing as well.

The paper's core mechanism is easy to follow from the architectural diagram (Figure 2) and the main equation (Equation 3). The qualitative examples provide concrete evidence that the model is learning the intended structure. E.g., Figure 1 shows a high-level "marriage" feature with "divorce" and "engagement" subfeatures.

**Weaknesses:**

One limitation is that proposed architecture implements a two-level hierarchy (parent/child). Real-world semantics are often deeper. The paper doesn't discuss the limitations of this two-level structure or how the architecture might be extended to model deeper, more complex hierarchies.

The subspace dimension $s$ is a key hyperparameter, set to 4 or 8. While this is motivated by the low-rank finding from prior work and benefits efficiency, there is no sensitivity analysis or further justification for these specific values. It's unclear how performance would change with a larger $s$.

Last but not least, since feature absorption is a central problem that the paper tackles, adding references to prior sota/influential work on this benchmark, such as https://scholar.google.com/citations?view_op=view_citation&hl=en&user=ycscpaQAAAAJ&citation_for_view=ycscpaQAAAAJ:2osOgNQ5qMEC and https://www.alignmentforum.org/posts/zbebxYCqsryPALh8C/matryoshka-sparse-autoencoders, would better contextualize the contribution.

**Questions:**

Continuing from the weaknesses, H-SAE implements a two-level hierarchy. Have you considered or explored extending this to deeper, multi-level hierarchies (e.g., by making the low-level "expert" SAEs hierarchical themselves)? What challenges do you think will likely arises (e.g., optimization stability, vanishing activations)?

Could you clarify the impact of the $\mathcal{L}_{ortho}$ term in the main experiments? The appendix ablation suggests it's mainly for preventing dead latents. Does it have any negative or positive impact on the quality of the learned hierarchy or the reconstruction performance on the residual streams, compared to just using a different dead-atom mitigation technique (like the one from Gao et al.)?

---

> ### Author Response · Authors · 2025-11-30
>
> Thank you for your review and support\! Here’s a few responses to your central questions
>
> 1. (additional levels of hierarchy) Previous work shows that a level of hierarchy can be represented without \*any\* explicit structure through bi-orthogonality (Park et al.). For example, flat SAE could in principle learn distinct features for animal, dog, corgi, etc. We only make one level of the hierarchy explicit, but the H-SAE retains the ability to model arbitrary additional levels in the same manner as the flat SAE. The advantages of making the hierarchy explicit are mainly computational (via the MoE structure) and that it groups related features together (via the hierarchical structure). Even just making one level of the hierarchy explicit has a dramatic effect here. Additional levels of explicit hierarchy could be put into place using the same strategy, but we expect actually learning the structure effectively would require an enormous amount of data.
> 2. (subspace dim) The subspace dimension can really only vary from 4-64 to maintain the computational benefits of low rank structure. On the word ablations in the appendix, we find that performance is not sensitive to the projection dimension\!
>    1. This really connects to our broader perspective on the parameters that could be tuned. We think that something like the whitening operation we use for the words is needed on the residual stream, and see that as a key future direction. Until we get the geometry of the space we’re learning in “right” we saw it as a secondary goal to aggressively tune all our hyperparameters, but still worth reporting for reproducibility.
> 3. (orthogonality penalty) The inspiration for the orthogonality penalty are results from Park et al showing that bi-orthogonality of representations relates closely to semantic free variability. The hope was that imposing this term would improve the compositionality of the features. However, the ablations on words (see appendix) showed no obvious benefit for interpretability (though it’s unclear if it’s simply because compositionality is hard to measure\!). However, we observed empirically that this term is highly effective at preventing dead atoms, and is simpler and cheaper than the Gao et al approach, so we continued to use it. However, we do not claim it as a main contribution of the paper.

---

### Official Review · Reviewer_H1X7 · 2025-11-01

**Soundness:** 3
**Presentation:** 2
**Contribution:** 3
**Rating:** 4
**Confidence:** 4

**Summary:**

This paper proposes a novel SAE architecture that models learned features in an explicit hierarchy. Specifically, it employs a mixture-of-experts method that activates sub-features within activated features of the model. The authors empirically explore how the architecture improves the “reconstruction-interpretability” tradeoff, finding that HSAEs improve reconstruction and feature absorption.

**Strengths:**

- Hierarchical encoding is a very interesting topic and an important component of modeling LLM representations. Hierarchical structure is inherently quite interpretable, and intuitively resolves many issues with existing SAEs (such as splitting and absorption).
- Additionally, this work is grounded in existing literature on the hierarchy of representations in language models, and the MoE architecture is an intuitive operationalization of hierarchies.
- The results are promising, particularly for efficiency and for absorption.

**Weaknesses:**

- Given that for each high-level feature, there are 16 or 64 sub-features, shouldn’t the authors  be comparing to standard SAEs with 16x or 64x the number of normal latents (essentially the size of all of the sub-features from HSAEs combined) to ensure overall width is the same? If they are already doing this, please ignore this comment but clarify this point in the paper.
- Why not always compare to both TopK and Matryoshka SAEs? For each experiment, the authors only use one or the other as the baseline, making it difficult to determine where HSAEs improve upon current baselines.
- The interpretability gains are not currently very clear to me. Given the qualitative examples provided, hard to tell which is better (e.g. Figure 4, App D.2). Is there a way to qualitatively measure improvement? While automated interpretability is known to have limitations, results on this metric would be helpful. Similarly, evaluations of downstream interpretability applications (such as steering) would also help back of the claims of improved interpretability.

- Minor comments:
    - Please move Figures 3 and 4 closer to their references in the text.
    - Please provide standard deviations for Figure 5b.
    - The captions of Figures 3 and 5 are not within the margin requirements, please add them to the normal caption with the tags (a) and (b) inline.

**Questions:**

- To measure compositionality, please see Archetypal SAE proposed metrics.
- Did the authors explore any examples of improved downstream utility for HSAEs over existing SAE architectures?
- The implementation of hierarchical SAEs feels very similar to the notion of sparse group sparsity from prior sparse coding and dictionary learning literature. Did the authors try the optimization techniques from prior works on group sparsity?
- What is the motivation for the “small ℓ1 sparsity penalty on the latent values on both the top and low-level features outside the top k”? Won’t this simply result in downscaling?

[1] Fel, Thomas, Ekdeep Singh Lubana, Jacob S. Prince, Matthew Kowal, Victor Boutin, Isabel Papadimitriou, Binxu Wang, Martin Wattenberg, Demba Ba, and Talia Konkle. "Archetypal sae: Adaptive and stable dictionary learning for concept extraction in large vision models." ICML (2025).

---

> ### Author Response · Authors · 2025-11-30
>
> Thank you for your review\! We’re glad you found the grounding clear and our results promising. With respect to the concerns about scalability & comparisons:
>
> 1. (fairness of comparisons) In traditional flat SAEs, total width, computational complexity, and the number of parameters all scale the same. A main advantage of the hierarchical structure is that these three things decouple. In fact, because of the MoE and projected subspace structure, in our actual experiments the additional computational cost and number of parameters added to account for the hierarchical structure is comparatively tiny. For example, in the experimental setting with k=32, proj\_dim=8, top\_level\_dim=2304, number\_of\_low\_level\_atoms\_per\_expert=64, and number of top level atoms 8k, the standard SAE has \~18.9 million active parameters, and the H-SAE adds on only 600,000. That is, moving from the standard SAE to the H-SAE increases parameter count by just 3%. In all other experimental settings, this increase is even smaller. (Changing the x-axis of the plots to be this value is virtually imperceptible). Despite this tiny increase, this model outperforms a 32k standard SAE with 113M more parameters on reconstruction: we think this decoupling is really powerful\!
>
> 2. (Comparison with M-SAE) The improvement of the H-SAE relative to baselines is just that 1\. Incorporating hierarchical structure dramatically improves computational efficacy, and 2\. by explicitly organizing the concepts hierarchically, they are easier to parse (i.e., even if the H-SAE, standard SAE, and Matrysoka all found “dog” and “corgi” features, only the H-SAE is capable of making explicit that the two are connected.) Importantly, these advantages are \*by definition of the architecture\*. What the experiments test is, essentially, whether the training actually works. In fact, the M-SAE changes are orthogonal to ours and both techniques could be combined\!
>    1. For computational gains, the relevant baseline is really the SAE, because it gives a sense of the scale of the ‘free lunch’ improvement from the modified architecture. (Note that it’s known already that M-SAEs are both more expensive to train than baseline SAEs and also have worse reconstruction error for the same number of latents).
>    2. Given the structure of our mode, for interpretability, the questions we’re actually trying to answer (1) are the hierarchical arrangements discovered basically coherent (2) do we get any backsliding on non-hierarchical interpretability? M-SAEs are more expensive to train than Top-K, so we only do comparisons on the word level to make sure we don’t have any backsliding as compared to an existing way of increasing interpretability, and find this to be true
>    3. The ‘Chicago’ example on the appendix is worth looking at carefully. Since we have an explicit organization of features, we can observe when a low-level feature activates alongside a high level feature. For feature 175 and its sublatent 19 we get a “US states” top feature and “midwestern states” low level feature. In comparison, the M-SAE (1) learns less general features (having a feature for exactly the token chicago (2) learns less lower quality features (even the token level feature is split into 2, and we observe seemingly meaningless features that are active)
> 3. (interpretability measures) We agree that it’s not 100% clear how to measure interpretability quantitatively, but a key advantage of our structural change is we know we have hierarchically interpretable features by design so long as the model basically works. As an alternative to more misleading quantitative measurements we worked on enabling qualitative evaluation without concerns of cherrypicking. To that end, we included notebooks where you can do your own visualization and sets of randomly selected feature activations to explore\!
> 4. (l1 sparsity, group sparsity) We normalize our features at each step (as is standard for SAEs) to avoid scaling issues. The l1 sparsity then encourages features outside the top-k to not activate strongly (as to avoid small meaningless activations). Group sparsity on the other hand doesn’t have as clear applicability to our setting. Our explicit hierarchy doesn’t cleanly transfer ideas from group LASSO. So, using top-k and explicit gating seems to us to be the correct approach. But this is worth investigating\!

---

### Author Response · Authors · 2025-12-04

Thank you to all reviewers for their detailed comments.

All reviewers appreciated the importance of incorporating hierarchical semantics into SAE-type methods and found the hierarchical mixture-of-experts approach used here to be clear and compelling. Many reviewers also highlight the experimental evaluation and results as strengths.

There were two main concerns raised.
1. Fairness of comparison (H1X7, Y3qo, iFYM).  Reviewers noted that we compare baseline SAEs and H-SAEs by the number of top-level atoms (instead of total atoms) and worry this is unfair. As we explain in detail in the individual responses, our choice here is downstream of the fact that the H-SAE structure decouples the number of atoms, the number of parameters, and the computational cost (in standard SAEs these are all essentially the same). Essentially, the number of top level atoms dominates number of parameters (always >=97%) and computational cost (the hierarchy overhead is negligible) so our plots should be understood as a fair comparison of total number of parameters / total compute cost.

2. Interpretability evaluation and comparison to Matrysoka SAEs. Reviewers were concerned about quantitative evaluation of interpretability and, in particular, comparison to Matrysoka SAEs as a baseline. Here, the main point of clarification is that the advantages of the H-SAE structure relative to the baseline---computational improvement and explicit grouping of semantically related features---is *by design* and not an emergent property of the training. Accordingly, the main experimental evaluation is simply showing that training works without degradation relative to baselines. We also included additional interpretability experiments on the top-level atoms as sanity checks for model behavior. It's nice that these checks show improvements in compositionality measures in the way we would expect, but we emphasize that this is closer to a bonus than the core point of the paper, and the contribution would be strong either way. With respect to Matrysoka in particular, we include comparisons in the body of the paper on the token-level for an interpretability comparison to another method designed to increase interpretability. Our response to reviewer H1X7 contains a detailed analysis of this comparison, demonstrating how the H-SAE learns high quality features. But, at a high level: The M-SAE changes are orthogonal to ours and both techniques could be combined!

We believe that we have fully clarified these concerns in the discussion with the reviewers. Additionally, we trained 5 new H-SAEs for 200M tokens (scaling across sparsity, suggested by iFYM) finding that H-SAEs scale with sparsity favorably (as expected). Because of this year's ICLR snafu it is not possible to have the usual back and forth, but we are confident that the concerns are fully addressed and the reviewers would have raised scores.

---

### Meta-Review · Area_Chair_Er4u · 2026-01-06

**Summary:**

This paper proposes a novel SAE architecture that models learned features explicitly in a hierarchy. Specifically, it employs a mixture-of-experts method that activates sub-features within the model's activated features. The authors empirically explore how the architecture improves the “reconstruction-interpretability” tradeoff, finding that HSAEs improve reconstruction and feature absorption.

The reviewers raised two main categories of concerns about the paper. First, multiple reviewers questioned the fairness of comparisons, noting that H-SAEs appeared to have many more parameters than baseline SAEs due to the expert sub-SAEs (16x or 64x more), and worried this made performance gains misleading rather than demonstrating genuine architectural advantages. Second, reviewers were concerned about insufficient evaluation of interpretability claims and inadequate comparison to Matryoshka SAEs, which were relegated to the appendix despite being the most closely related work. Several reviewers specifically requested more comprehensive quantitative interpretability metrics, sparsity-scaling experiments across multiple levels (as recommended by SAEBench), and sensitivity analyses for key hyperparameters, such as subspace dimension. Additionally, reviewers noted that the qualitative examples provided were insufficient to validate the core hierarchical learning claims and suggested the paper needed deeper investigation into what hierarchical structures were actually learned and whether the method could extend beyond two levels of hierarchy. The most critical reviewer argued these evaluation gaps were severe enough to recommend rejection, while others felt the limitations placed the work marginally below the acceptance threshold despite acknowledging the promising core idea. The authors responded to the concerns mostly by arguing why the comparisons do not make sense in their setting; however, after reading the rebuttal and the reviewers' comments, I think their paper could have done more empirically to showcase the proposed architecture, with more ablations and comparisons (see below for more detail).

**Reviewer Concerns:**

After carefully reading the paper, reviewer comments, and author response,  I believe the authors' arguments are largely sound, but they could have been more proactive with ablations/experiments. I summarize the concerns below:

1. Fair comparison justification (parameter count)

The authors explain  that H-SAEs add only 3% more parameters (600K on top of 18.9M) while achieving better performance than models with 113M more parameters. The decoupling of total width, computational cost, and parameters is a legitimate architectural advantage, not unfair comparison.

2. Design vs. emergent properties

The authors argue that hierarchical structure and computational gains are "by design" rather than emergent. The main empirical question is whether training works without degradation, not whether hierarchy magically emerges.

3. Sparsity conceptualization challenge

The authors identify that sparsity becomes conceptually harder with forced co-activation in hierarchical structures. Their point about treating "married pairs" of features together makes sense given the architecture.

Some concerns were not fully/adequately addressed. For example, for the late sparsity scaling experiments, the authors trained only 5 new H-SAEs across sparsity levels AFTER reviewer feedback. This is a critical ablation that should have been in the original submission, especially since SAEBench explicitly recommends evaluating across sparsity ranges. Multiple reviewers noted that Matryoshka-SAE comparisons were relegated to appendix with limited scope (word-level only, no explained variance comparison). Given M-SAE is the most closely related work, this deserves main text treatment with comprehensive metrics.

The authors' defense that "M-SAE changes are orthogonal and could be combined" is somewhat evasive, in my view, If methods are complementary, demonstrate that explicitly.

While the authors emphasize qualitative evaluation tools (notebooks, random samples) to avoid cherry-picking, quantitative interpretability metrics were sparse. A reviewer specifically suggested Archetypal SAE metrics for compositionality. The authors' response that "compositionality is hard to measure" comes across as somewhat dismissive.

The absorption experiments are good, but a more systematic interpretability evaluation (e.g., automated interpretability scores and downstream task performance, such as steering) would strengthen the claims.

A reviewer also asked about subspace dimension sensitivity (why 4 or 8?). The authors claimed  "performance is not sensitive" based on word ablations, but didn't show residual stream ablations. This feels incomplete given it's described as a "key hyperparameter."

Finally, multiple reviewers asked about extending beyond a 2-level hierarchy. The authors claim that additional levels would require "enormous amounts of data," but they provided no empirical evidence or pilot experiments to support this claim. A small-scale pilot would have been valuable.

**Reviewer Scores:**

I am not sure these reviewers would have changed their scores, given the provided *additional* experiments were scarce.

---

### Decision · Program_Chairs · 2026-01-26

Reject